# Programmable nanocomposites of cellulose nanocrystals and zwitterionic hydrogels for soft robotics

Rasool Nasseri [1,7], Negin Bouzari[1,7], Junting Huang[1], Hossein Golzar[2], Sarah Jankhani[1], Xiaowu (Shirley) Tang [2,3], Tizazu H. Mekonnen[1,4,5], Amirreza Aghakhani [6] & Hamed Shahsavan [1,3,4] ✉

Stimuli-responsive hydrogels have garnered significant attention as a versatile class of soft actuators. Introducing anisotropic properties, and shape-change programmability to responsive hydrogels promises a host of opportunities in the development of soft robots. Herein we report the synthesis of pH-responsive hydrogel nanocomposites with predetermined microstructural anisotropy, shape-transformation, and self-healing. Our hydrogel nanocomposites are largely composed of zwitterionic monomers and asymmetric cellulose nanocrystals. While the zwitterionic nature of the network imparts both self-healing and cytocompatibility to our hydrogel nanocomposites, the shear-induced alignment of cellulose nanocrystals renders their anisotropic swelling and mechanical properties. Thanks to the self-healing properties, we utilized a cut-and-paste approach to program reversible, and complex deformation into our hydrogels. As a proof-of-concept, we demonstrated the transport of light cargo using tethered and untethered soft robots made from our hydrogels. We believe the proposed material system introduce a powerful toolbox for the development of future generations of biomedical soft robots.

Small-scale soft robots, in the past decade, have introduced unorthodox yet promising strategies for minimally invasive medical interventions such as biopsy, single-cell transport, and targeted drug delivery in confined biological environments[1,2]. The challenging nature of the fabrication of small-scale soft robots and minimizing the use of traditionally rigid and bulky onboard powering, sensing, and actuation components have drawn the attention of soft robotic researchers to stimuli-responsive soft materials with inherently integrated sensing, actuating, and powering capabilities. Indeed, the majority of lately reported small-scale soft robots are essentially soft constructs from stimuli-responsive materials that can move, morph, and perform a set of dynamic functions[3–7].

To operate in real-world environments, soft robots should be capable of complex shape deformation by sensing and generating adaptive responses to different stimuli in unpredictable situations[8,9]. Employing stimuli-responsive materials in soft robotics requires the ability to program their shape-morphing, typically rendered by an anisotropic assembly of their microstructures, to elicit desirable functions[10]. The functionality and applicability of such robots in biomedical applications largely depend on their biocompatibility, desired interaction with biomolecules such as proteins, and biodegradation[11]. Moreover, the capability of self-repair is highly desirable for soft robots which are inherently vulnerable to mechanical damage in dynamic and evolving surroundings[12]. The design and fabrication of a

[1]Department of Chemical Engineering, University of Waterloo, Waterloo, ON N2L 3G1, Canada. [2]Department of Chemistry, University of Waterloo, Waterloo, ON N2L 3G1, Canada. [3]Centre for Bioengineering and Biotechnology, University of Waterloo, Waterloo, ON N2L 3G1, Canada. [4]Waterloo Institute for Nanotechnology, University of Waterloo, Waterloo, ON N2L 3G1, Canada. [5]Institute for Polymer Research, University of Waterloo, Waterloo, ON N2L 3G1, Canada. [6]Institute of Biomaterials and Biomolecular Systems (IBBS), University of Stuttgart, Pfaffenwaldring 57, 70569 Stuttgart, Germany. [7]These authors contributed equally: Rasool Nasseri, Negin Bouzari. ✉e-mail: hshahsav@uwaterloo.ca

small-scale soft robot with all the mentioned attributes is still a challenging task.

Among the large number of soft materials known to soft roboticists, shape-memory hydrogels, and hydrogel actuators are particularly promising in the design of miniaturized bioinspired soft robots[13–19]. In hydrogel actuators, spatial swelling/shrinking in the aqueous environment in response to an external stimulus manifests in the form of macroscale shape morphing[16]. A well-known method to program shape-morphing is to introduce anisotropy into the hydrogel microstructure to induce a differential swelling. Local internal stresses arising from such differential swelling of hydrogels result in anisotropic shape transformations such as twisting, bending, or folding[17,18]. The directed assembly of nanofillers, void channels, or liquid crystal domains in pre-polymerized hydrogel precursors followed by their in-situ polymerization is an effective approach to induce anisotropic microstructures in hydrogels[20–25]. Unidirectional shear flow, exposure to external magnetic and electric fields, and directional freezing are some of the conventional methods for the directional assembly of asymmetric nanofillers, void channels, or the creation of aligned liquid crystalline domains in hydrogels[21,25–29].

Biomedical hydrogel soft robots are naturally susceptible to being fouled by biomolecules and prone to trigger foreign body reactions (FBR)[30,31]. Aside from the health concerns they cause, biofouling and FBR disrupt the intended functionality of soft robots. Zwitterionic hydrogels are known for their superior anti-fouling properties and biocompatibility[32,33]. They possess super hydrophilicity, zero net charges, and H-bond accepting functional groups, which leads to minimal protein adsorption and cell adhesion[32]. These characteristics make them great candidates for designing miniaturized medical soft robots with minimal FBR. To the best of our knowledge, anti-fouling zwitterionic hydrogels have never been used as stimuli-triggered shape-morphing materials, especially for soft robotic applications.

In this work, we have synthesized hydrogel nanocomposites, based on zwitterionic monomers and cellulose nanocrystal (CNC) nanoparticles with promising potential for use in small-scale soft robotics. Structural anisotropy promoted by the alignment of CNC enables our hydrogels to undergo programmable stimuli-responsive shape transformation. By employing a cut-and-paste strategy owing to the self-healing properties of this hydrogel we were able to prepare complex stimuli-triggered shape-morphing systems. A proof-of-concept robotic functionality was demonstrated by the combination of programmable stimuli-responsiveness, and a cut-and-paste approach in designing tethered and untethered soft grippers. Further studies on the cytotoxicity of our hydrogels revealed their high level of cytocompatibility. Our results offer a new perspective on the design of adaptable hydrogel nanocomposite and its programming strategy for the development of tethered and untethered small-scale biomedical soft robots.

## Results
### System Concept
To synthesize self-healing hydrogels that undergo programmable stimuli-responsive shape transformation, we designed a synthetic protocol based on the copolymerization of 3-dimethyl (methacryloyloxyethyl) ammonium propanesulfonate (DMAPS) and methacrylic acid (MAA) in the presence of CNC nanoparticles. It is known that the copolymerization of sulfobetaine methacrylates as zwitterionic monomers with a small amount of MAA improves the mechanical properties of the resulting hydrogel due to the hydrophobic associations imposed by $CH_3$ groups of MAA[34]. The zwitterionic hydrogels from a random copolymer of DMAPS, and MAA have shown self-healing properties as a result of the dynamic electrostatic interactions between the oppositely charged groups and hydrogen bonding[30,35–38]. Additionally, P(DMAPS-MAA) copolymers

are proven sensitive to temperature, ionic strength, and pH, making them great building blocks for stimuli-responsive shape-morphing soft robots[39–43].

Cellulose nanocrystals are outstanding examples of 1-D nano-reinforcer materials which have gained increasing attention due to their sustainable sources, unique mechanical properties, and surface characteristics[44]. Similar to other 1-D nanoparticles with high aspect ratios, confined CNC nanoparticles could be oriented by external shear forces. This strategy is versatile and compatible with extrusion 3D printing and has been previously used to induce anisotropy in hydrogels[25,45–47]. Inspired by this strategy, the shear-induced alignment of CNCs in our P(DMAPS-MAA) hydrogel precursors could be utilized to induce structural anisotropy (Fig. 1a, b). The self-healing properties of P(DMAPS-MAA) would also enable us to employ a cut-and-paste strategy to prepare complex stimuli-triggered shape-morphing systems (Fig. 1c). A combination of programmable stimuli-responsiveness and a cut-and-paste approach could be used to fabricate both tethered and untethered soft grippers (Fig. 1d, e).

To find the soft actuator with optimal self-healing and mechanical properties, responsiveness, and programmability, we prepared a series of hydrogel precursors with different chemical compositions, which were then polymerized between two glass slides separated with spacers of known thickness under UV light at a wavelength of 365 nm. In this systematic work, we synthesized and studied physically-crosslinked hydrogels first, then modified their formulation by chemical crosslinking, and finally added CNC nanoparticles to achieve programmable soft actuators from hydrogel nanocomposite with desired properties.

### Physically-crosslinked hydrogels
We synthesized hydrogels without any chemical crosslinkers (or physically crosslinked) and systematically studied their self-healing and mechanical properties. In all precursors, DMAPS and MAA were the main comonomers polymerized with different weight ratios ranging from 1:2 to 4:1 (DMAPS:MAA). Results of the tensile test in Fig. 2a demonstrate that physically crosslinked hydrogel with a 3:1 DMAPS:MAA weight ratio (called GelWC hereafter), although soft, is still mechanically robust and practical for soft robotic applications. The tensile strength and elongation at the break of these hydrogels are presented in Supplementary Fig. 1a, b. In tandem, strips of physically crosslinked hydrogel samples were cut in half and the two pieces were brought in contact for self-healing experiments. First, we studied the effect of time on the self-healing of GelWC and noticed a gradual enhancement of healing efficiency with time. After about 4 h GelWC showed full recovery, Fig. 2b. The comparison between the results of mechanical testing on original physically crosslinked hydrogels of different formulation and their self-healed counterparts after 6 h also revealed the complete recovery of mechanical properties, of the hydrogel containing at least 75 wt% of DMAPS with respect to the total monomer weight (Fig. 2c). The similarity of the elastic modulus of soft robots and soft body tissues is essential for the non-invasive interaction of them with the body as well as the manipulation of soft objects, like tissue cells[48–50]. The elastic modulus of our hydrogel is around 30 kPa which is in the range of some soft body tissues (such as Uterus tissue ⌣2–250 kPa[51,52] and Liver tissue ⌣10 kPa[53]) and smaller than some others (such as Kidney tissue ⌣90–180 kPa[54])and Small intestinal tissue ⌣2500–5500 kPa[55]) ensuring the safe and non-invasive interactions. Given its excellent self-healing and acceptable mechanical properties, the composition of GelWC was chosen for further modification in our study.

Response to external stimuli, such as pH, in the form of reversible shape-morphing triggered by swelling/deswelling, is crucial for the functionality of hydrogels as soft robots. To confirm the stimuli-responsiveness of GelWC samples, we studied their swelling/deswelling mechanism in response to changes in environmental pH, which is

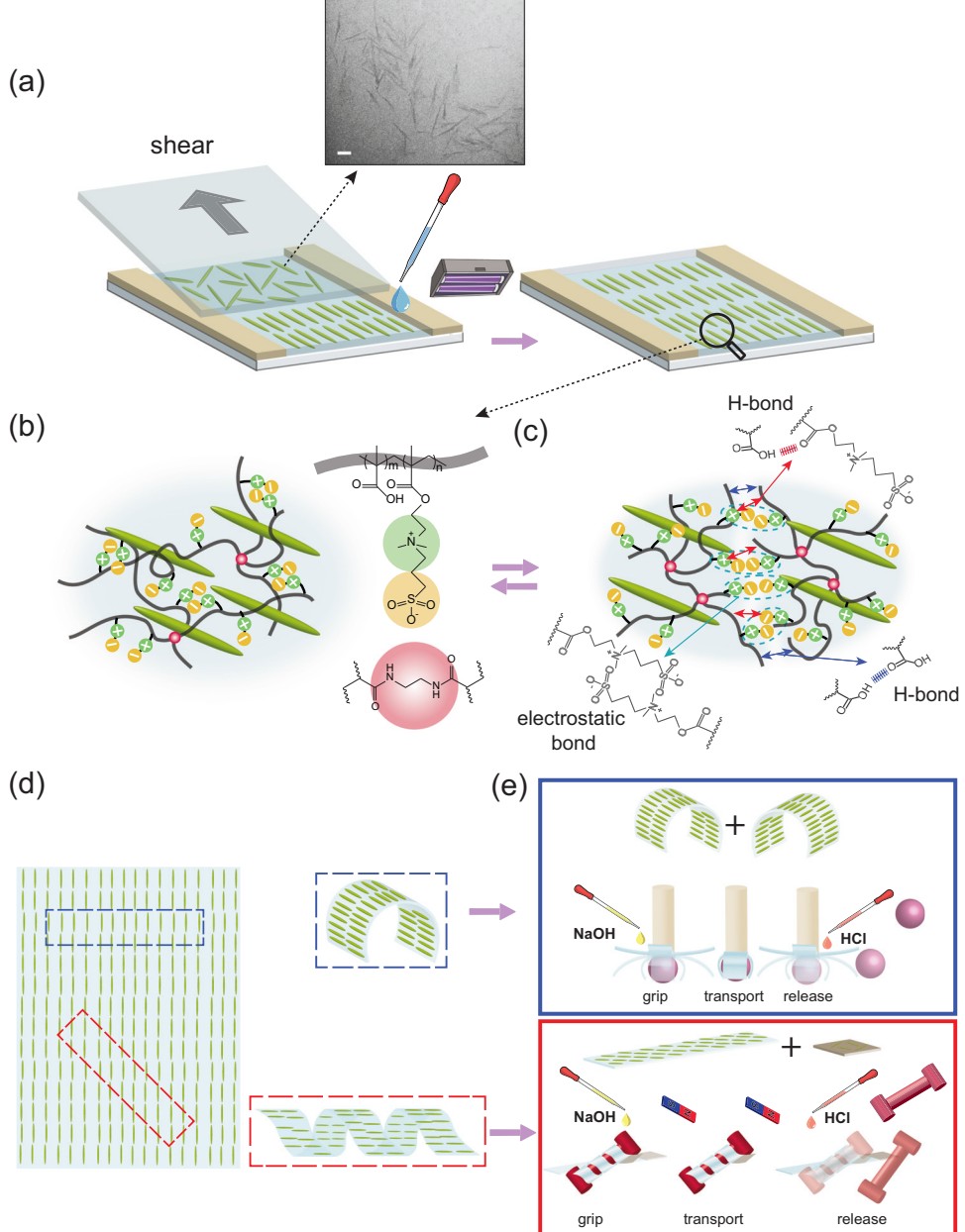

**Fig. 1 | System Concept. a** The schematic of the fabrication of the anisotropic stimuli-responsive hydrogel and TEM image of CNC nanoparticles. The scale bar is 100 nm. **b** The chemical structure of the hydrogel components. **c** Schematic of the self-healing mechanism of the hydrogel through noncovalent reversible crosslinking. **d** Shape-change programming of the anisotropic hydrogel. **e** Schematic of the cut-and-paste strategy to design functional tethered (blue box) and untethered (red box) soft grippers. By adding a magnetic patch to the untethered soft gripper, it can be navigated and steered by an external magnetic field.

schematically depicted in Fig. 2d. Figure 2e shows the degree of swelling ($(L - L_0)/L_0$, where $L_0$ and $L$ are the initial and the post-swelling lengths of samples, respectively) of a GelWC sample in neutral, high, and low pHs. After about 2 h samples reached an equilibrium in water (pH ~ 7). By placing the samples in a buffer with pH 12 they started to swell rapidly. Placing the samples in a buffer with pH 3 forced them to lose some water and come back to their original size. To confirm the effect of pH on the swelling/deswelling behaviour of the GelWC samples after equilibration in water, they were first placed in a buffer with pH 3 and then in a buffer with pH 12. The size of the samples did not change at pH 3 but increased rapidly at pH 12 (Supplementary Fig. 1d). At high pH values (greater than 4.7), the –COOH groups of MAA are ionized, and the charged –COO$^-$ groups repel each other, leading to the swelling of the hydrogel while at lower pHs this process is reversed[39]. However, the swelling/deswelling of GelWC

samples with changes in the pH was not repeatable and the hydrogel started to degrade after the first cycle (Supplementary Fig. 2a). This can be most likely attributed to the breaking of the dipole-dipole attraction and intergroup electrostatic association due to the screening of zwitterionic moieties by oppositely charged ions in the environment[56].

## Chemically-crosslinked hydrogels

To alleviate the reversibility issue of GelWC swelling/deswelling, we added a minuscule amount of N, N′-methylenebis(acrylamide) (BIS) as a chemical crosslinker to its formulation. Using this approach, we anticipated introducing chemical crosslinking to our hydrogels would enhance the reversibility of pH-responsive swelling/deswelling. After a systematic variation of BIS concentration, along with swelling/deswelling experiments, we found that hydrogels with 167:1 weight ratio of

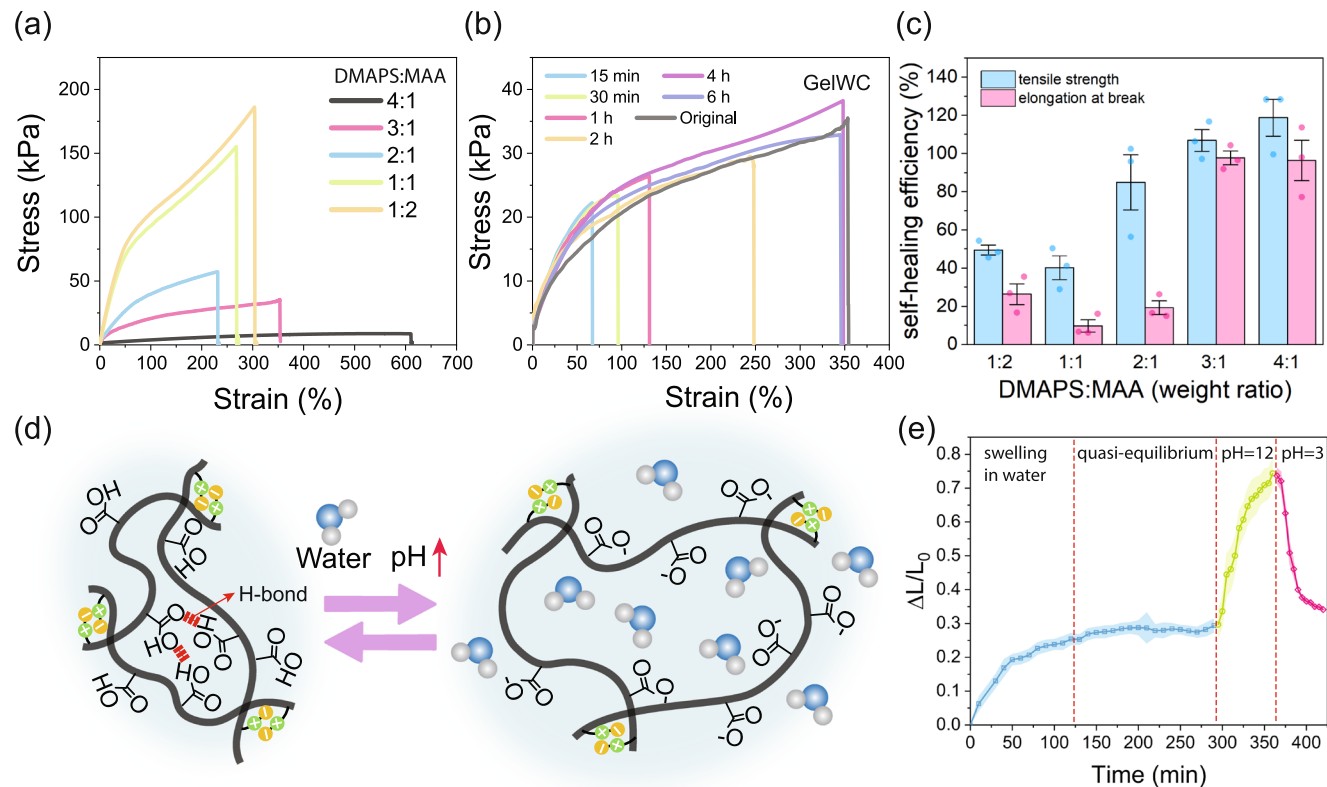

**Fig. 2 | Properties of physically-crosslinked DMAPS-MAA hydrogels. a** Stress-strain behavior of physically-crosslinked DMAPS-MAA hydrogels containing different ratios of comonomers. Three samples were tested, and the best representative of all samples is shown in the graph. **b** Stress-strain behavior of GelWC after self-healing as a function of time. Three samples were tested, and the best representative of all samples is shown in the graph. **c** Self-healing efficiency of physically-crosslinked DMAPS-MAA hydrogels samples containing different ratios of comonomers after 6 h of healing. The average value of three samples is reported. The error bars represent the standard error. **d** Schematic presentation of the swelling/deswelling mechanisms of a GelWC sample in response to pH. At high pH values (greater than 4.7), the −COOH groups of MAA are ionized, and the charged −COO⁻ groups repel each other, leading to the swelling of the hydrogel while at lower pHs this process is reversed. **e** The degree of swelling of a GelWC sample in water (pH - 7) and in buffers with pH 12 and 3. The average value of three samples is reported. The shaded area represents the standard error.

comonomers:BIS (molar ratio 143:1) show reversible swelling/deswelling, which are called Gel hereafter (Supplementary Fig. 2b). Supplementary Fig. 1c shows a schematic representation of the swelling/deswelling mechanism in chemically crosslinked Gel samples upon a change in pH.

## Hydrogel nanocomposites

Inducing anisotropy into the microstructure of hydrogels has been proven to lead to differential swelling and is an essential step in the fabrication of programmable shape-morphing hydrogels[19,57–59]. The unidirectional shear of 1D nanoparticles, such as CNC, has been employed in a few reports as an effective method of inducing anisotropy to hydrogels' microstructure[23,60,61]. It is also known that CNC nanoparticles can be aligned by external stress induced by shear, magnetic, and electric fields, especially when they are present in a liquid crystalline phase[28,62,63]. As such, we postulated that the addition of CNC nanoparticles to hydrogel precursors can impart a liquid crystalline phase and their shear alignment can lead to anisotropy in our DMAPS-MAA hydrogels microstructure. For this, we modified the formulation of the Gel precursor and replaced water with a 10 wt% CNC suspension (see materials and methods for details). In our preliminary experiments, we found that the concentration of CNC in the final hydrogel precursor must be above 4 wt% to ensure the realization of the liquid crystalline phase. Furthermore, we confirmed the shear thinning of Gel precursors containing CNC nanoparticles in rheology tests. The results in Supplementary Fig. 4, show that CNC nanoparticles can

reorient and align upon application of shear. No shear thinning was observed for Gel precursor without CNC which confirms the role of CNC and its orientation in the shear thinning behavior of the precursor. Gel precursors containing 5 wt% CNC nanoparticles were accordingly polymerized in capillary cells either to anisotropic hydrogels with preferential CNC alignment (AGel) or isotropic counterparts without any preferential CNC alignment (IGel). AGel samples were prepared by casting the precursor on a glass substrate and confined by spacers, followed by applying a unidirectional shear force via moving a glass countersubstrate on top several times (Fig. 1a). To prepare IGel samples the countersubstrate was gently placed on the casted precursor without any lateral movement. The photopolymerization was then carried out to fix either the unidirectional or random alignment of the CNCs inside hydrogel precursors.

We performed FTIR tests to confirm that the presence of CNC does not disturb the polymerization reaction (Supplementary Fig. 2). The MAA spectrum showed a distinctive peak at approximately 1635 cm⁻¹, corresponding to the stretching of the C = C[64]. The peak at around 1630 cm⁻¹ is attributed to the C = C stretching in the structure of DMAPS[65]. After polymerization the amplitude of the C = C characteristic peak (highlighted with blue) reduced substantially in AGel, IGel, and Gel spectra, suggesting the successful polymerization of the monomers[66]. The small peak at 1060 cm⁻¹ in the IGel and AGel spectra (highlighted with green) is due to the stretching of C-O bonds of CNC as it can also be observed in the CNC spectrum[67].

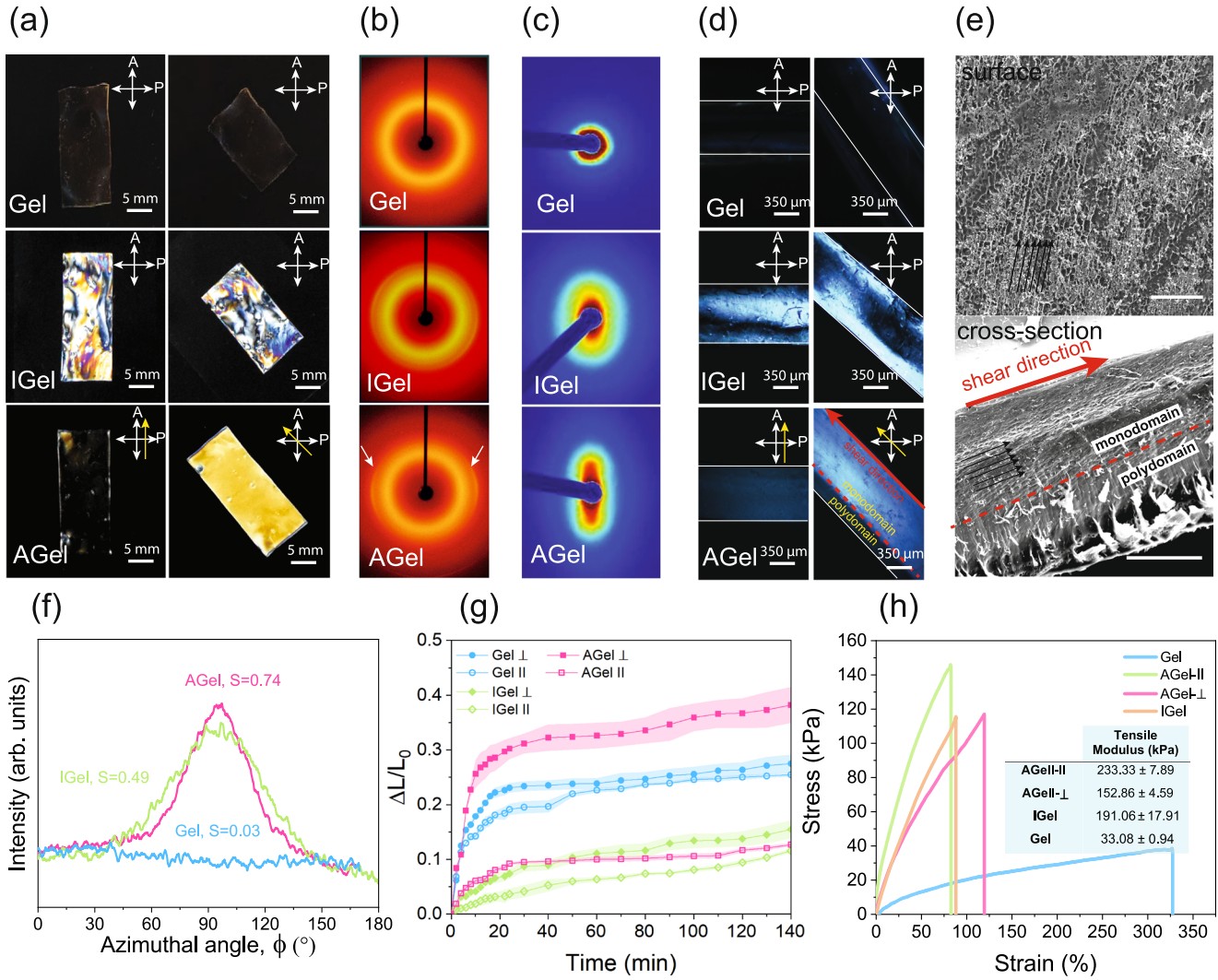

**Fig. 3 | Anisotropy of hydrogel nanocomposites. a** Images of Gel, IGel, and AGel between crossed-polarizers. CNC alignment direction is shown with a yellow arrow. **b** 2D-XRD patterns of Gel, IGel, and AGel. **c** 2D-SAXS patterns of Gel, IGel, and AGel. **d** POM images of the cross-section of Gel, IGel, and AGel. CNC alignment direction is shown with a yellow arrow. The red dashed line separates the regions with alignment (monodomain) and without alignment (polydomain). **e** SEM images of AGel surface exposed to shear and cross-section. Black arrows show the direction of CNC alignment. The red dashed line separates the regions with alignment (monodomain) and without alignment (polydomain). The size of the error bar is 100 µm and the initial thickness of the sample before freeze-drying was 700 µm. **f** The intensity with respect to the azimuthal angle at $2\theta = 22.9°$ for Gel, IGel, and AGel. **g** Degree of swelling vs time for Gel and IGel in two perpendicular directions and that of AGel parallel and perpendicular to the CNC alignment. The average value of three samples is reported. The shaded area represents the standard error. **h** Tensile test results of Gel, IGel, and AGel in parallel and perpendicular directions to CNC alignment along with the average tensile modulus ± standard error calculated using three samples. Three samples were tested, and the best representative of all samples is shown in the graph.

## Anisotropy of hydrogel nanocomposites

Different techniques were employed to confirm the anisotropy in the hydrogel microstructure. Hydrogels containing CNC (IGel and AGel) are transparent under natural light but show vivid interference colors when viewed between crossed polarizers or by a polarized optical microscope (POM) (Fig. 3a and Supplementary Fig. 5). No interference colors were observed in the images of Gel taken between crossed polarizers or with a POM that confirms the role of CNCs in the interference colors. Regardless of the orientation with respect to POM's polarizer/analyzer, IGel displays a heterogeneous multicolor transmission pattern confirming the isotropic microstructure of the hydrogel whose precursor was polymerized in a polydomain liquid crystalline phase. In contrast, AGel shows a uniform monochromic transmission when observed between crossed polarizers. The transmission through the sample viewed between crossed polarizers (or by POM) is the highest when the longer side of the sample (parallel to the shear) is positioned at 45° with respect to the polarization axis and the

sample appears dark when it is parallel to any of the polarizer or the analyzer. These results suggest that the CNCs present in hydrogel precursors indeed form a monodomain liquid crystalline phase that is aligned along the shearing direction and retained even after the photopolymerization[68].

The orientation of CNCs in hydrogels was also evaluated by two-dimensional X-ray diffraction (2D-XRD). For Gel the diffraction pattern was completely circular and angle independent indicating the amorphous structure of the material (Fig. 3b). A rather weak and slightly angel-dependent diffraction pattern at all azimuthal angles on the diffraction at $2\theta = 22.9°$ (corresponding to the (200) diffraction of cellulose $I\beta$ crystal in CNCs) was observed for IGel which suggests that CNCs are parallel in the film but randomly oriented in all directions (polydomain) (Fig. 3b). The diffraction pattern of AGel, however, showed equatorial arcs and strong angle dependence, revealing the unidirectional alignment of CNCs along the shearing direction (monodomain) (Fig. 3b). The intensity with respect to the azimuthal

angle, $I(\phi)$, on the (200) scattering plane also confirmed the alignment of CNCs along the shearing direction in AGel (Fig. 3f). To quantify the extent of alignment, Herman's order parameter ($S$) was calculated based on the diffraction intensities at $2\theta = 22.9°$ (details presented in Calculation of Herman's Order Parameter section, Supplementary Information). The values of $S$ are 0.03, 0.49, and 0.74 for Gel, IGel, and AGel, respectively, suggesting the isotropic structure of Gel and a higher degree of anisotropy in the plane of the film for AGel compared to IGel[69]. We also performed two-dimensional small-angle x-ray scattering (2D-SAXS) on AGel, IGel, and Gel. For Gel the diffraction pattern was completely circular and angle-independent. The diffraction pattern of IGel was oval-shaped and slightly angle-dependent and for AGel it was completely elongated which further confirmed the unidirectional alignment of CNCs along the shearing direction (Fig. 3c). Supplementary Fig. 5d shows the 1D radial-averaged SAXS plots of Gel, IGel, and AGel. The low-q curves show a $\sim q^{-1}$ asymptote, which is because the length of CNC is much larger than its width. At large-q values, all profiles exhibit a $\sim q^{-4}$ asymptote, which can be explained by the presence of a sharp interface[70].

To investigate the uniformity of the alignment along the thickness the cross-section of AGel (parallel to the shear) was studied by POM. POM image of the AGel cross-section positioned 45° with respect to the polarization axis revealed two distinct layers one bright and one dark confirming the anisotropy gradient along the thickness (Fig. 3d). The sample appears dark when it is parallel to the polarization axis. The POM images of the IGel cross-section positioned 45° and parallel to the polarization axis appeared similar confirming polydomain microstructure (Fig. 3d). These images confirm that moving a glass slide on the top surface of hydrogel precursors produces a shear stress gradient along the thickness. However, the applied shear is not large enough to induce the alignment of CNCs in the precursor throughout the thickness beyond a certain threshold leading to a gradient in anisotropy. The self-assembly of CNCs close to the bottom glass substrate can induce alignment and anisotropy. However, the domain size (or thickness) of such self-assembled structures is very small. We have done SEM on the cross-section of IGel and as it can be seen in Supplementary Fig. 6b the thickness of this layer is very small compared to the rest of the film which has layered structures with periodic spacing and spiral stacking of CNCs that are characteristic of chiral nematic assemblies. As such surface aligned domains alone do not play a tangible effect on the overall anisotropy.

To study the effect of applied shear on the AGel samples' microstructure, scanning electron microscopy (SEM) images were taken from their surface and cross-section (Fig. 3e)[44]. The SEM image of the AGel surface exposed to the shear before polymerization revealed the microstructural alignment in the direction of the shear while SEM images from its cross-section confirm the POM results showing an anisotropy gradient along the thickness. A typical AGel sample indeed shows less pronounced unidirectional microstructural anisotropy for regions closer to the bottom substrate. The larger portion of aligned thickness in the POM image of the AGel cross section is likely because POM images were taken from swollen samples and the higher swelling of the aligned region compared to the isotropic region. The gradient of microstructural anisotropy along the thickness can be observed even after reducing the sample thickness to 250 μm (Supplementary Fig. 6a). To compare the microstructure, the SEM image of the cross-section of IGel with 500 μm is presented in Supplementary Fig. 6b. Layered structures closer to the substrate farther from the shear resemble randomly oriented domains of chiral nematic CNC aggregates reported in the literature[69]. Further investigation is required to identify the exact nature of the phase and alignment.

We then studied the effect of CNC alignment on the degree of swelling of Gel, IGel, and AGel hydrogels in water (Fig. 3g). All hydrogels start to swell upon immersing in water and after about 1 hr reach a pseudo-equilibrium state in which their swelling rate decreases

significantly. The expansion of Gel in all directions was relatively the same, however, the expansion of IGel in the direction perpendicular to the long axis of the cast hydrogel is slightly more than that of the parallel direction to the long axis suggesting that the casting process induced minor anisotropy to the hydrogel. This observation is consistent with 2D-XRD and 2D-SAXS results. For AGel the swelling in the direction of alignment is much more pronounced than that of the parallel direction to the alignment direction. Such a considerable in-plane differential swelling is shown in Supplementary Movie 1. By increasing the pH or ionic strength of the environment, the anisotropic swelling of AGel becomes even larger and more noticeable.

To study the effect of CNC nanoparticles addition and their alignment on the mechanical properties of the hydrogel at micro and macroscales, we performed rheology and tensile tests on dimensionally identical circular and rectangular specimens of Gel, IGel, and AGel, respectively.

For rheology, we cut circular discs with identical radii and thickness from Gel, IGel, and AGel samples. Supplementary Fig. 7 demonstrates the rheological behavior of these hydrogels. To confirm the formation of hydrogels at room temperature the frequency-dependent dynamic shear moduli measurements were performed in the linear viscoelastic region (1% strain, 1 Hz) to ensure that the microstructure of the hydrogels was preserved. In the entire frequency range, the storage modulus G′ was greater than the loss modulus G″ suggesting the solid-like behavior of all hydrogels (Supplementary Fig. 7a)[71]. The storage modulus of Gel was the highest among the three hydrogels followed by IGel and AGel in the entire frequency range. The decrease in the storage modulus of IGel and AGel can be attributed to the drastic enhancement of hydrophilicity and water content of hydrogels by introducing CNC to their formulation, as shown in Supplementary Fig. 9. The strain amplitude sweep test on the hydrogels indicated that the 1% strain at which the frequency-dependent dynamic shear moduli measurements were performed was in the linear viscoelastic region of the hydrogels (Supplementary Fig. 7b)[72].

For the tensile test, we cut specimens from Gel and IGel samples whose long axis were either parallel or perpendicular to the long axis of the hydrogels. We also cut specimens from an AGel sample, whose long axis was parallel, and perpendicular to the shearing direction. As anticipated, Gel specimens showed almost identical mechanical properties measured in different directions (Supplementary Fig. 8). For IGel, the tensile modulus and tensile strength were slightly higher in the parallel direction compared to the perpendicular direction which confirms minor induced anisotropy during the casting (Supplementary Fig. 8). AGel showed higher tensile modulus and tensile strength in both directions compared to Gel (Fig. 3h). The tensile modulus and the tensile strength of AGel specimens stretched parallel to the alignment direction are higher than those of AGel specimens stretched perpendicular to the alignment direction. Such a large difference in mechanical properties of Gel and IGel, and those of AGel measured parallel and perpendicular to the alignment indicates the key role the presence and alignment of CNC nanoparticles play in the reinforcement of IGel and inducing mechanical anisotropy in AGel hydrogel composites, respectively. In the linear viscoelastic range for which the storage modulus of Gel is higher than AGel, the deformations are minuscule so short-range electrostatic interactions as the source of physical crosslinking can stay intact and the plasticization of water molecules is the main reason for the difference between mechanical properties of different samples. In tensile tests, however, the deformations are large so electrostatic interactions are disrupted. As a result, CNCs as nano-reinforcers play a more significant role. That is the reason for the higher tensile modulus of hydrogels containing CNC compared to Gel.

We then investigated the influence of CNC alignment on the optical properties of AGel subject to elongation. Supplementary Fig. 5e shows the gradual color changes for the specimens of AGel stretched

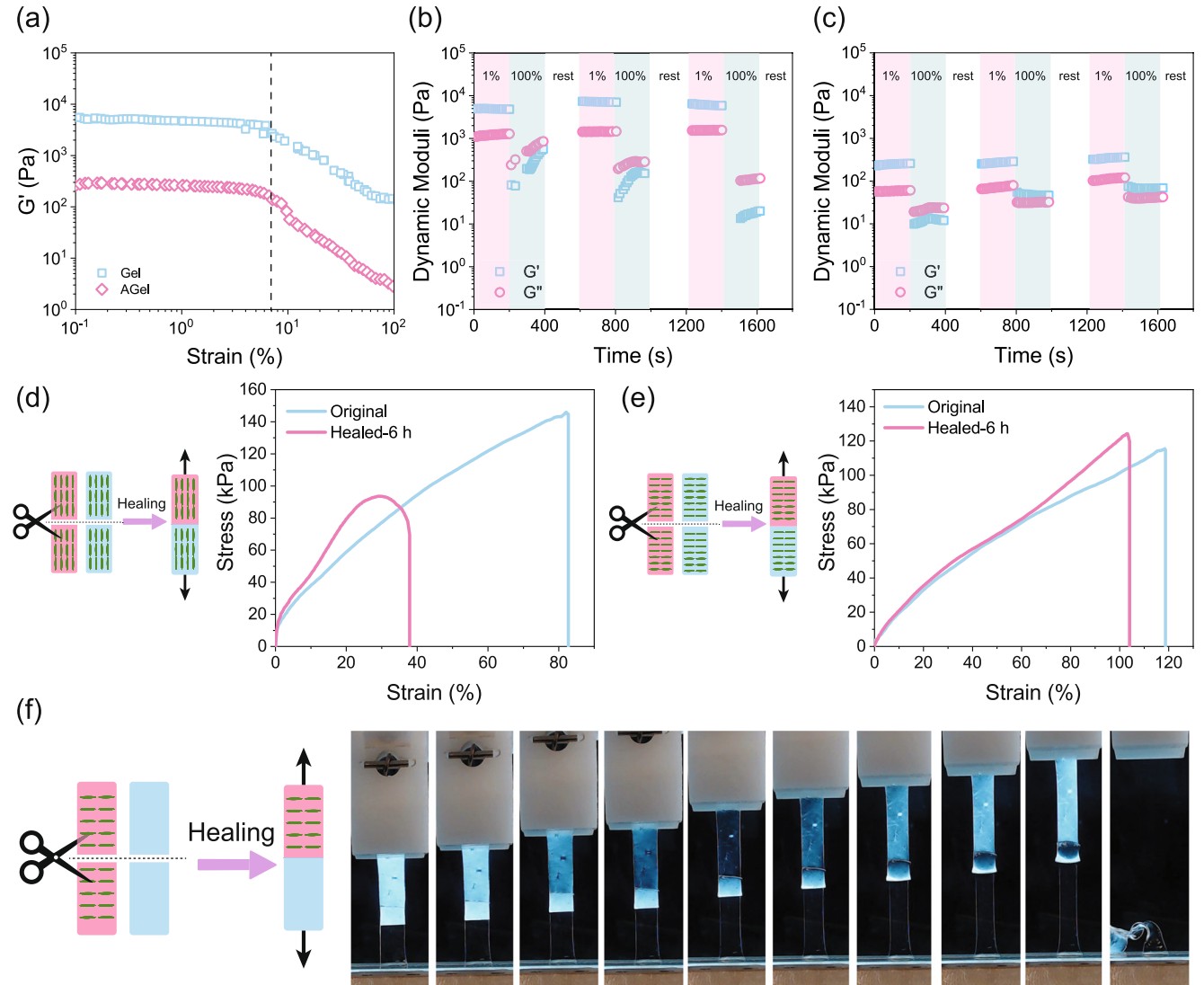

**Fig. 4 | Self-healing of hydrogel nanocomposites. a** Amplitude sweep results on Gel and AGel showing the linear viscoelastic region. Step-strain experiment on **b** Gel and **c** AGel. Large strain (100%) dropped the values of G′ (squares) and G″ (circles), indicating breakage of crosslinks. G′ and G″ were recovered under a small strain (1%). Self-healing of AGel in **d** alignment and **e** perpendicular directions after 6 h. Three samples were tested, and the best representative of all samples is shown in the graph. **f** Stretching a rectangular sample (5 mm × 25 mm) made of one piece of Gel (lower half) and one piece of AGel cut in the perpendicular direction (upper half) healed for 6 h. Due to the rotation of CNCs upon extension, the upper half becomes first transparent and then white again.

parallel and perpendicular to the direction of CNC alignment. The color change pattern is completely different for these two specimens confirming the anisotropy in the microstructure. The color change for AGel stretched parallel to the alignment direction is mostly due to the change of specimen thickness while the CNCs alignment is maintained intact[68]. For AGel stretched perpendicular to the alignment direction, elongation of the specimen at first disrupts the orientation of CNCs leading to the loss of birefringence while with further elongation, CNCs start to orient in the perpendicular direction and causing birefringence again[28].

## Self-healing of hydrogel nanocomposites

To realize the cut-and-paste strategy to prepare complex stimuli-triggered shape-morphing constructs, having proper self-healing properties is very crucial. Between the two major approaches of dynamic covalent crosslinking and noncovalent physical crosslinking in self-healing hydrogels[73], our hydrogel falls under the second category. Dynamic hydrogen bonds and ionic associations facilitate the interdiffusion of the chains across the crack and their engagement in the new junctions (Fig. 1c)[74]. To evaluate the recovery of hydrogel

mechanical properties at the microscale following the network rupture at high strain, step-strain measurements were performed[75]. When Gel and AGel were subjected to a small strain in the linear viscoelastic region (1%, Fig. 4a) for 200 s, G′ was greater than G″. By increasing the strain to a value greater than the linear viscoelastic threshold (100%, Fig. 4a) for 200 s, both G′ and G″ dropped immediately indicating the network disruption. After 200 s rest, the strain was reduced to 1% again, hydrogel exhibited complete recovery of both G′ and G″. Alignment of CNC seems to help preserve the solid-like properties of the AGel under high strains in the second and third cycles probably due to the higher available surface area as a result of fewer contacts and more effective interactions with the matrix. IGel showed similar behavior in the step-strain experiment (Supplementary Fig. 7c). This result was reproducible upon additional strain cycles as can be seen in Fig. 4b, c. To evaluate the self-healing efficiency of hydrogels at the macroscale, tensile tests were performed on the original and self-healed samples. The self-healing efficiency of the Gel was evaluated after 6 h of healing and limited self-healing was observed. In the presence of the crosslinker, the ability of the hydrogel to retain water decreased. The expelled water accumulated on the surface of the Gel

samples interrupted the self-healing process. Adding CNC to the hydrogel enhances the hydrophilicity of the network which can prevent water accumulation on the surface of the hydrogel by retaining the water inside. Consequently, healing efficiency of ~50% and ~90% was observed for AGel in alignment and perpendicular directions, respectively (Fig. 4d, e). Higher swelling of AGel in the perpendicular direction compared to the alignment direction (Fig. 3e) facilitates the chain movements and interdiffusion across the crack resulting in more effective healing in this direction. We also examined the self-healing of Gel and AGel samples with isotropic and anisotropic swelling profiles. Figure 4f shows a rectangle (5 mm × 25 mm) made of one piece of Gel (lower half) and one piece of AGel cut in the perpendicular direction (upper half) after 6 h of healing. After around 100% strain, the sample broke but not at the healed area indicating that the healing of AGel and Gel pieces was effective and that the healed region was at least as strong as the pristine hydrogels, Supplementary Movie 2.

## Shape-change programming

The reversible in-plane and out-of-plane anisotropic swelling triggered by pH, the gradient of microstructural anisotropy along the thickness, and the initial geometry of single-layer AGel samples provide a robust set of adjustable parameters to program the 2D-to-3D shape-morphing. The gradient in microstructural anisotropy along the thickness facilitates bending in a mechanism similar to classical bimorph hydrogels subject to nonidentical swelling along their thickness[76–79]. The unidirectional alignment of CNC on the regions closer to the top surface can guide such bending deformation in preferential directions by decoupling the deformation characteristics from the geometry of the initially flat hydrogel[80]. Our system can be deemed as a bimorph with nonidentical swelling behavior along the thickness in which CNCs are oriented randomly on one side, with a characteristic thickness of $h_I$ (~200 μm), and unidirectionally aligned on the other side, with a characteristic thickness of $h_A$ (~600 μm) (Fig. 5a). Similar to the deformation profile of typical bistable bimorphs, straining of the layer with the unidirectionally aligned CNCs (with higher swelling) is constrained by the layer with randomly oriented CNCs (with lower swelling). Therefore, while one layer is loaded with tension, the other layer experiences compressive stresses and to release those internal stresses, the generated bending moment results in the bending of the flat hydrogel[81]. We have modeled the deformation behavior of the hydrogel bimorphs with finite element simulations. Through anisotropic swelling, if the long axis of strips is either perpendicular or parallel to the dominant swelling direction, the simulations show that the strip will undergo pure bending (Fig. 5b). Deviation of the strips' long axis orientation from the dominant swelling direction, in turn, leads to helicity as a result of the balance between forces induced by elastic stretching and bending (Fig. 5b)[82–84].

To verify our simulations, we cut different samples from an AGel film whose long axis was perpendicular, parallel, and 45° with respect to CNC alignment to investigate how the CNC alignment affects the shape-morphing of the hydrogel. To trigger the shape change we first increased the pH of the hydrogel environment (to ~12) and to recover the original shape we decreased the pH (to ~3). Figure 5c shows two strips of AGel in which the CNC alignment is perpendicular and parallel to their long axis, respectively. Both strips showed reversible bending, one parallel and the other perpendicular to their long axis (Supplementary Movie 3 and Supplementary Movie 4). The bending direction in both strips was dictated by the larger swelling which occurs perpendicular to the CNC alignment. When the angle between the CNC alignment and the long axis of the strip is 45°, the hydrogel twists and forms a helix (Fig. 5c, Supplementary Movie 5)[85]. The pitch of the helix can be adjusted by the angle at which the strip is cut with respect to the CNC alignment[82–84]. The time scale of both shape deformation and recovery was around 5 min. It is also possible to trigger the shape change in AGel strips by swelling them in a saline solution.

Supplementary Fig. 10 shows the pictures of the shape changes of AGel pieces in response to ionic strength in a 2 M NaCl solution after 5 mins. The reversibility in shape change and soft actuators' durability are major requirements for the functionality of soft robots over a desirable time. Therefore, we also tested these parameters for AGel strips. Supplementary Fig. 2d shows the reversible bending and unbending of a short strip of AGel for 7 cycles in response to pH. The bending angle was defined as the angle between a vertical tangent applied to one edge of the hydrogel and a vector connecting two edges of the hydrogel after bending[76].

The predictable swelling behaviors of Gel and AGel single layers, as well as their self-healing characteristics enable the creation of a host of responsive constructs with complex deformation profiles using a cut-and-paste approach. Figure 5d shows three examples of 2D constructs made by cut-and-paste of a few pieces of single-layer Gel or AGel samples with different CNC alignment and their expected shape change. Figure 5e shows the POM images of assembled multi-piece constructs from Gel or AGel samples and CNC alignment in them. We predicted the overall shape-change of multi-piece constraints using finite element simulations (Fig. 5f). Figure 5g and Supplementary Movie 6–8 show complex reversible 3D shape morphing of the hydrogel construct triggered by increasing and decreasing the pH. In accordance with our prediction, the anisotropic regions bend upon swelling, while the isotropic regions exhibited uniform swelling. The reversibility of the actuation allows the 3D shapes to return to their flat configuration by decreasing the pH (Supplementary Movie 6–8). The time scale of both shape deformation and recovery was around 5 min.

## Cell viability and degradation of hydrogel nanocomposites

In order to examine the applicability of fabricated hydrogel in biomedical soft robotics, we investigated the cytotoxicity properties of AGel, IGel, and Gel. To study the effect of the comonomer ratio, we also prepared a hydrogel with the filliped weight ratio of comonomers (DMAPAS:MAA weight ratio 1:3). To study the cytocompatibility of the hydrogels, a live/dead assay was performed, and cell proliferation was monitored by fluorescence microscopy over 5 days (Fig. 6a). For all the hydrogels, cell viability was very high (above 95%) over the 5 days of incubation alongside the hydrogels, confirming their biocompatibility. The hydrogel prepared with the filliped weight ratio of comonomers showed high toxicity leading to the death and detachment of most of the cells on day one of the experiments confirming the effect of zwitterionic monomer on the biocompatibility of the hydrogel (Supplementary Fig. 11a). On-demand degradation of hydrogel is essential for its sustainability (no waste at the end of its lifetime) and biomedical application (biodegradation in the body). In an aqueous salt solution, the ions break the dipole-dipole attraction and intergroup association between the zwitterionic moieties thus promoting the swelling and finally degradation of the hydrogel, Fig. 6c[56]. Although the hydrogel without crosslinker (GelWC) is stable for a long time underwater, it started to dissolve fast in 10 wt% NaCl solution upon immersion (Fig. 6b and Supplementary Movie 9). In this work, chemically-crosslinked hydrogels are not fully degradable (Supplementary Fig. 11b). Albeit we anticipate that using a degradable crosslinker such as zwitterionic carboxybetaine disulfide cross-linker makes them completely degradable[86].

## Effect of physiological temperature on the hydrogel nanocomposite

We evaluated the shape change of AGel pieces cut with the CNC alignment perpendicular to the long axis and making a 45° angle with the long axis in response to pH change at temperatures close to the physiological temperature. Due to the upper critical solution temperature (UCST) nature of thermo-responsivity of the hydrogel nanocomposite[34], increasing the temperature, increased the speed of shape change (Supplementary Fig. 12a and Supplementary Movie 13).

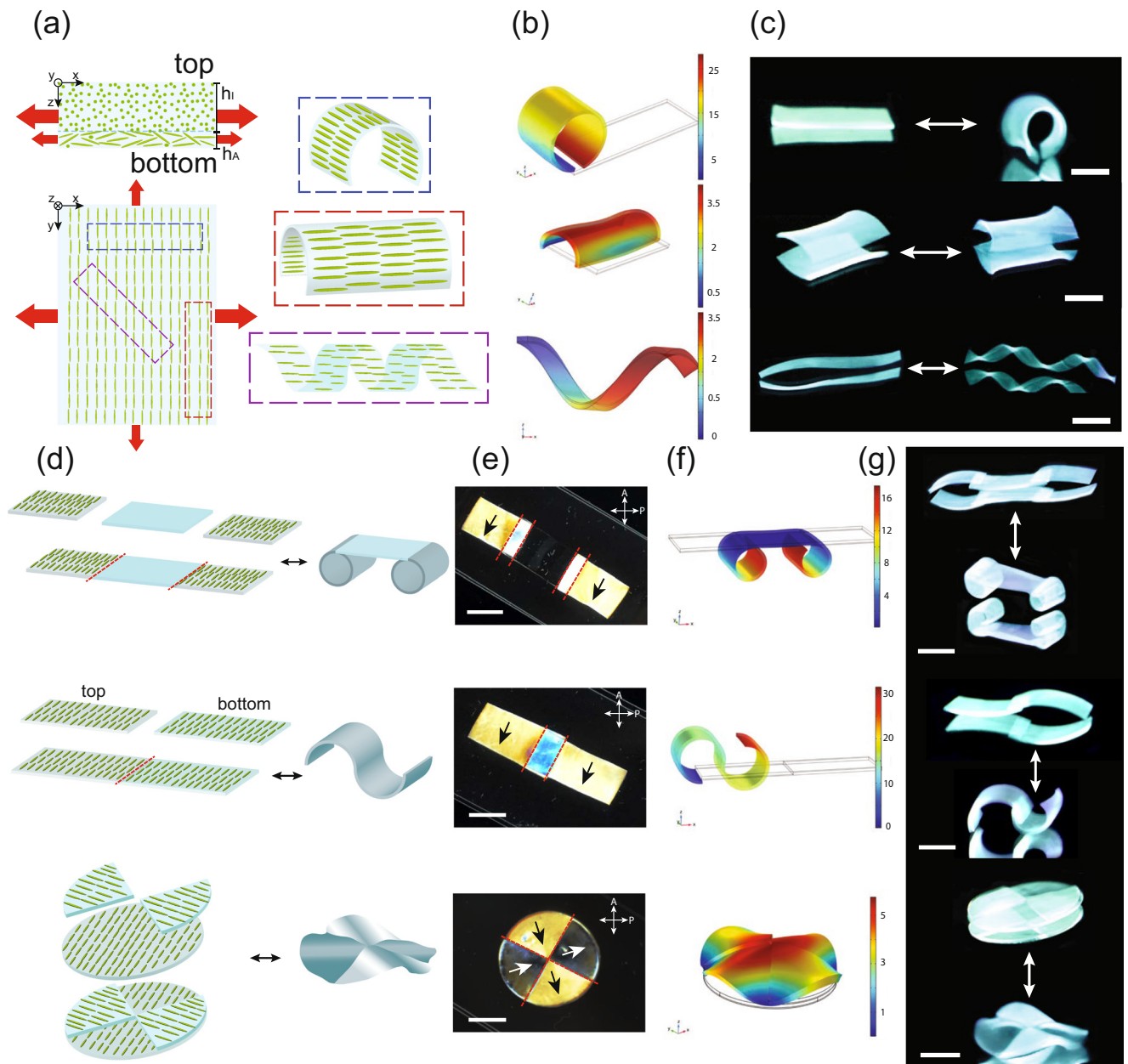

**Fig. 5 | Shape-change programming. a** Effect of CNC alignment with respect to the long axis of the cut piece in the shape morphing of the piece after swelling. We cut AGel film at different angles to obtain perpendicular (blue box), parallel (red box), and 45° (purple box) CNC alignment with respect to the long axis of the cut piece. Our system can be deemed as a bimorph with nonidentical swelling behavior along the thickness in which CNCs are oriented randomly on one side, with a characteristic thickness of $h_I$ (~200 μm), and unidirectionally aligned on the other side, with a characteristic thickness of $h_A$ (~600 μm). The size of the red arrows shows the level of swelling in different directions. **b** Results of finite element simulations of shape change of AGel pieces cut with the CNC alignment perpendicular to the long axis (top), parallel to the long axis (middle), and making a 45° angle with the long axis (bottom) in response to pH change. Blue-to-red gradients show the total displacements in mm. **c** Reversible shape change of AGel pieces cut with the CNC alignment perpendicular to the long axis (top), parallel to the long axis (middle), and making a 45° angle with the long axis (bottom) in response to pH change. The scale bars are 10 mm. **d** The cut-and-paste approach to assemble the AGel and Gel pieces to make a construct with complex shape-morphing. **e** Images of the assembled AGel and Gel pieces to make constructs with complex shape-morphing between two perpendicular polarizers. The scale bars are 10 mm. Self-healed regions are shown by a red dashed line. CNC alignment is shown by black and wight arrows. **f** Results of finite element simulations of hydrogel constructs in response to pH change. Blue-to-red gradients show the total displacements in mm. **g** Reversible shape changes of hydrogel constructs in response to pH change. The scale bars are 10 mm.

The time scale of both shape deformations was around 30 sec. To investigate the effect of physiological temperature on the mechanical properties of the hydrogel, we performed temperature-sweep and strain-sweep tests on AGel. By increasing the temperature, both G′ and G″ increased with a slight step at UCST around 57 °C (Supplementary Fig. 12b). The linear viscoelastic region of the hydrogel also slightly increased (Supplementary Fig. 12c). To study the self-healing properties of hydrogel nanocomposite, a step-strain experiment was performed on AGel at 37 °C (Supplementary Fig. 12d). By increasing the strain to a value greater than the linear viscoelastic threshold (100%, Supplementary Fig. 12b) for 200 s, both G′ and G″ dropped immediately indicating the network disruption. After 200 s rest, the strain was reduced to 1% again, hydrogel exhibited complete recovery of both G′ and G″.

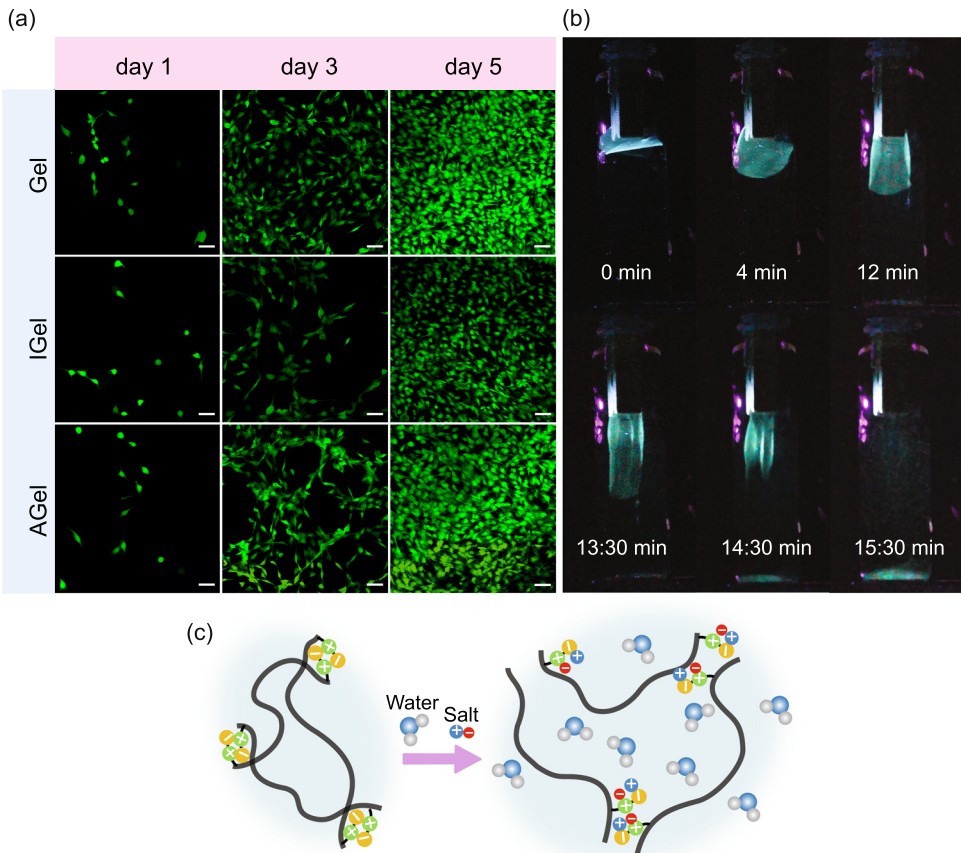

**Fig. 6 | Cell viability and degradation of hydrogel nanocomposites.**
**a** Proliferation of the incubated Fibroblast cells with Gel, IGel, and AGel samples monitored by fluorescent microscopy over 5 days. Scale bars are 50 μm. **b** On-demand degradation of GelWC in 10 wt% NaCl solution. **c** Schematic of the degradation of GelWC in a salt solution.

## Soft robotic application

After demonstrating programmable shape-morphing, self-healing, and biocompatibility of our hydrogel, we designed a pH-responsive miniature gripper as a proof-of-concept of its tethered robotic applications (Fig. 7a). The gripper was made by applying the cut-and-paste approach to attach two strips (5 mm × 30 mm) of AGel in which the direction of alignment was perpendicular to the long axis. The gripper was used to grab light spherical cargo by rolling the gripper arms around it when triggered by increasing the pH. After transferring the cargo to a new location, the gripper arms were opened by decreasing the pH, and the cargo was released (Fig. 7a and Supplementary Movie 10). Due to the softness of the gripper arms, they can also wrap around irregular and soft objects. In a separate experiment, we used our gripper to move a soft biological cargo with an irregular shape (chicken fat tissue), Supplementary Movie 11. Note that in both experiments, the hydrogel grippers prevent the lateral movement of soft and light cargo during transport and do not play a key role in load bearing in the normal direction against gravity. To test the applicability of our hydrogels in an untethered robotic application, we also fabricated a miniature robot that could be remotely navigated to transfer a very light cargo through a confined flooded space similar to a maze. In order to benefit from a smaller pitch, or more turns, in a helix, our untethered robot was made from a strip of AGel cut in the way that the angle between the CNC alignment and its long axis is around 60°. To grab the cargo soft robot twists around it at high pH. Lowering the pH results in opening the twists and releasing the cargo. We attached a patch of IGel prepared by magnetic CNC (MCNC) instead of CNC (IGelm) to the robot to enable remote navigation (Fig. 6b). MCNCs were prepared by co-precipitation of $Fe^{2+}$

and $Fe^{3+}$ ions on CNC (Supplementary Fig. 14a). MCNC nanoparticles exhibit a typical superparamagnetic behavior with extremely small hysteresis loops and coercivity (Supplementary Fig. 14b). IGelm showed a lower magnetization saturation compared to MCNC due to the dilution of the magnetic nanoparticles in the hydrogel (Supplementary Fig. 8b). This patch enables the magnetic navigation of the robot through the maze using a strong permanent magnet from outside of the workspace. When the robot reaches the confined walls, we increase the pH of the environment to twist the robot and grab the cargo. The robot can then be moved through the maze and finally release the cargo by returning to its original shape after increasing the pH, Fig. 7c and Supplementary Movie 12.

To evaluate the effect of pH on the mechanical properties of the soft robots, we have done a frequency-sweep on AGel after swelling in a solution with pH of 12 and 3 for about 5 min. Swelling at low pH did not change the dynamic moduli significantly compared to the sample swelled in water. Swelling at high pH, however, increased the dynamic moduli probably due to the stretch of chains as a result of extreme swelling (Supplementary Fig. 13).

## Discussion

In Summary, we fabricated a programmable shape-morphing hydrogel by shear-induced alignment of CNC nanoparticles in a zwitterionic stimuli-responsive hydrogel precursor, followed by photopolymerization under UV to lock the anisotropy in the microstructure. The hydrogel exhibited differential swelling and anisotropic mechanical properties essential for programable shape-morphing. Owing to the self-healing properties of hydrogel due to the dynamic hydrogen bonds and ionic associations, we were able to

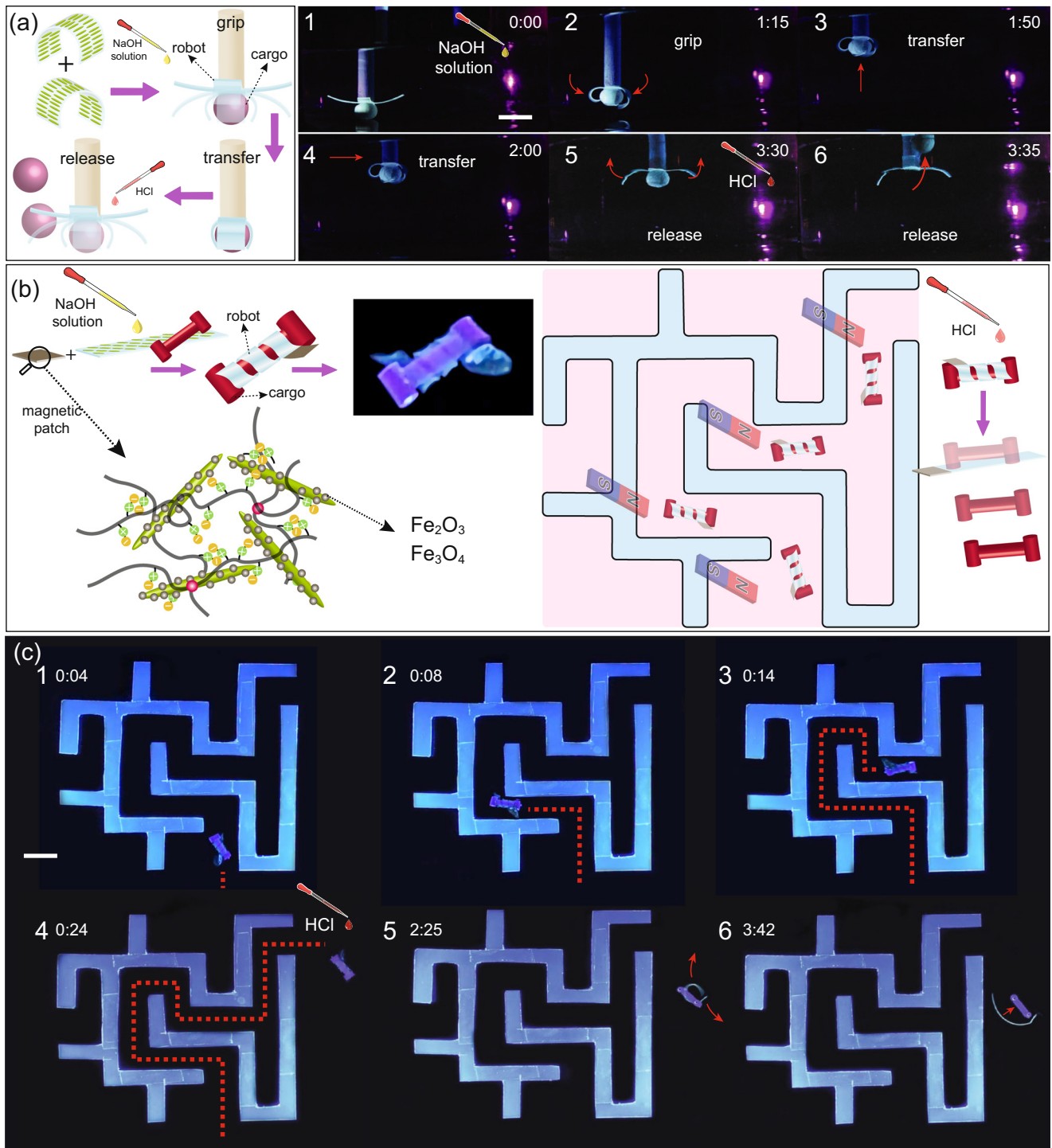

**Fig. 7 | Soft robotic applications of hydrogel nanocomposites. a** A micro-gripper developed by a cut-and-paste strategy that grabs a spherical cargo by rolling its arms triggered by increasing the pH. After transferring the cargo to a new location the gripper arms are opened and the cargo is released by decreasing the pH. Scale bar is 10 mm. **b** Schematic of a micro-robot transferring a light cargo by twisting around it triggered by increasing the pH. By adding a magnetic patch to the untethered soft gripper, it can be navigated and steered by an external magnetic field. **c** A micro-robot transferring an I-shaped cargo by twisting around it triggered by increasing the pH. The micro-robot is remotely navigated and steered by a magnet. Decreasing the pH triggers the opening of the micro-robot and the release of the cargo. Note that the cargo-grabbing process is not fully touchless and we had to direct the cargo to the right spot manually. Entirely touchless grasping step requires active control over the positioning of the twisting ribbon, which is not the scope of this work. The scale bar is 10 mm.

employ a cut-and-paste approach to fabricate reversible shape-morphing constructs with complex shape deformations. Investigation of the cytotoxicity and anti-fouling properties of our hydrogel revealed a high level of cytocompatibility making it a potential candidate for biomedical robotic applications. As proof of concept of its robotic applications, after demonstrating programmable shape-

morphing, self-healing, and biocompatibility of our hydrogel, we designed two pH-responsive small-scale robots. The first robot was a gripper capable of grabbing, transferring, and releasing spherical or irregular soft biological cargos triggered by pH. The second one was a robot capable of transferring light cargo by remote navigation and steering in confined environments using a magnetic field. The results

of this work can expand the development of adaptive and reconfigurable biomimetic soft robots.

There are fundamental issues that need to be addressed before our anisotropic programmable hydrogel is completely suitable for real-world applications. For instance, further work is required for the development of zwitterionic hydrogels with complex morphing and motion modes designed for advanced applications. Also, hydrogel formulations that provide enhanced mechanical properties could be more useful for higher load-bearing capacity. Furthermore, the miniaturization of our robotic contracts is an important challenge that lies ahead if they are aimed to be used in therapeutic or diagnostic settings. Adjusting our hydrogel to be triggered by milder pHs is another crucial factor for practical applications. In this work, we have investigated the shape-change properties in two extreme pH environments to expedite actuation. However, the pKa of the methacrylic acid dictates that the deformation of the system occurs at pH levels above 4.6 in principle. Further details are required to determine the exact pH profile that is effective in inducing shape changes. Our system holds the potential for therapeutic applications in body organs with high native pH levels, as well as the ability to tolerate acidic pH environments. The bladder serves as an example of such a favorable environment[87]. In addition, the durability of our hydrogel construct along with their on-demand degradability needs to be alleviated. To employ our hydrogel in a medium with a certain pH range in a real-world application, an elaborate study should be performed on the effect of its long-time exposure to that pH range. Finally, multifunctionality through designing hydrogel constructs with integrated functions of actuation, sensing, control, and communication is also a crucial step for future applications.

# Methods

## Materials

3-dimethyl (methacryloyloxyethyl) ammonium propanesulfonate (DMAPS, 95%), 2-Hydroxy-2-methylpropiophenone (97%), Iron(II) chloride ($FeCl_2$, 98%), Iron(III) chloride ($FeCl_3$, 97%), Hydrochloric acid (HCl, 37%), Sodium hydroxide pellets (NaOH, $\geq$ 97%), Ammonium hydroxide solution ($NH_4OH$, $\geq$ 30%), Ethanol ($> 98\%$), N,N′-Methylenebis(acrylamide) (BIS, 99%) and Methacrylic acid (MAA, 99%) were purchased from Sigma-Aldrich Co., Canada. Cellulose nanocrystal (CNC, $\geq$ 94%) was acquired from CelluForce Inc., Canada. Deionized water (DI water, $> 16$ M$\Omega$ cm resistivity) was used in all of the experiments. All the chemicals were used without further purification.

## DMAPS-MAA Hydrogel

Hydrogels were prepared by a one-step UV-polymerization of an aqueous solution of DMAPS and MAA monomers in the presence of BIS as a cross-linker. The total monomer concentration was kept at 50 wt%. In a typical batch, 3750 mg DMAPS, 1250 mg MAA (DMAPS:MAA weight ratio 3:1 and molar ratio 9:10), 30 mg BIS (monomer:BIS weight ratio 1000:6 and molar ratio 1000:7), and 30 mg 2-Hydroxy-2-methylpropiophenone (monomer:initiator weight ratio 1000:6 and molar ratio 1000:6.5) were added to 5 g of 10 wt% CNC suspension, while maintaining the total monomer concentration to 50 wt%. The hydrogel precursor was injected into a cell prepared by two glass slides (5 cm × 5 cm) separated by two strips of 500 μm spacer. Before UV-polymerization, the cell was placed in a preheated oven at 60 °C for 30 mins. UV-polymerization was conducted at 60 °C in a container purged with nitrogen. Each side of the cell was shined by UV light of wavelength 365 nm for 45 mins. After polymerization, hydrogels were washed with Milli-Q water to remove the residual unreacted chemicals.

## DMAPS-MAA-CNC Hydrogel Nanocomposite

CNC (10 wt%) was incorporated into the aqueous phase before the addition of other components. The amounts of monomers, cross-linker, and initiators were kept constant. The precursor was transferred on a glass slide between two strips of 500 μm spacer glued to the edges. The precursor was then sheared by the edge of another glass slide at a rate of 5 cm s$^{-1}$ a few times. The suspension was then covered with a glass slide. The rest of the procedure is similar to the previous section.

## Synthesis of Magnetic CNCs (MCNCs)

MCNC was synthesized using co-precipitation of iron salts on CNC as described elsewhere[68,72,88]. Briefly, the suspension of CNCs (100 ml, 0.2 wt.%) was degassed with $N_2$ for 20 min in a three-neck flask. After that, $FeCl_2$ (0.18 g) and $FeCl_3$ (0.47 g) were added to the mixture while stirred vigorously for an additional 30 min. The pH was adjusted by slowly adding ten drops of $NH_4OH$ to the suspension (pH-10) which caused it to turn black. After stirring for 30 min at 70 °C, the mixture was cooled to room temperature by adding deionized water. MCNC nanoparticles were washed repeatedly with distilled water and ethanol, then separated with an external magnetic field and centrifugation (4025 × g, Thermo Scientific™, Sorvall™ ST) to remove any impurities. The precipitated MCNC nanoparticles were then preserved in ethanol for further use.

## DMAPS-MAA-MCNC Hydrogel

MCNC (10 wt%) was incorporated into the aqueous phase before the addition of other components. The amounts of monomers, cross-linker, and initiators were kept constant. The precursor was transferred on a glass slide between two strips of 500 μm spacer glued to the edges. The suspension was then covered with another glass slide. The rest of the procedure is similar to the previous section.

## Characterization

Polarized optical microscope (POM) images were captured by Euromex iScope (Fullerscope) microscope. The morphology of CNC and synthesized MCNC nanoparticles was investigated by transmission electron microscopy. After removing ethanol and drying the MCNCs completely, the powder sample was dispersed in water (1 mg ml$^{-1}$), and about 10 μl of the dilute dispersion was cast onto the surface of a 300 mesh grid which was then dried at an ambient temperature for 24 hr before taking the images on a Philips CM10 transmission electron microscope. The rheological properties of the precursor and hydrogel were characterized using a rheometer (HAAKE MARS, Thermo Scientific) with a 20 mm parallel plate geometry and a gap of 0.7 mm. Viscosity vs shear rate was measured to confirm the shear thinning of the hydrogel precursor. The strain sweep (0.1–100%) was performed at a fixed frequency of 10 rad/s to determine the linear viscoelastic range. Then, a dynamic frequency sweep (0.1 to 100 rad s$^{-1}$) at a constant strain of 1.0% was applied. Water evaporation was minimized during the temperature sweep tests using a solvent trap. Finally, the step-strain test was conducted to investigate the self-healing properties of the hydrogel for several cycles. Tensile testing of hydrogel was performed using a CellScale tensile machine (UniVert, Canada). To evaluate the self-healing efficiency of hydrogels a strip with dimensions of 1 cm × 3 cm was cut. The sample was then cut from the middle and two halves overlapped 5 mm and slight pressure was applied to ensure full contact. After 6 hr tensile experiment was performed on the healed sample. The alignment of CNC nanoparticles was investigated by scanning electron microscopy (SEM). Hydrogels were frozen and snapped in the liquid nitrogen and dried in the freeze-dryer for 24 h. The final samples were mounted on a sample holder and gold-sputtered before taking the images with a Hitachi S-3500N VP SEM. Two-dimensional X-ray diffraction (2D-XRD) images were recorded in transmission mode with a Bruker Venture DUO equipped with a Photon III detector using a Cu Kα X-ray Microsource beam with a wavelength (λ) of 0.154184 nm at 1.10 mA, 50.00 kV for 300 s at 2θ = 0.00 and 100 mm from the sample to the detector. Small-angle X-ray

scattering (SAXS) patterns were collected with a photon energy of 12.18 keV and sample-to-detector distance of 2335 mm. SAXS patterns were collected in transmission geometry with a 10 s dwell time. Patterns were processed with GSAS-II. SAXS data was calibrated with silver behenate (AgBeh). SAXS patterns were integrated from q = 0.006 to q = 0.3 Å$^{-1}$. A Nicolet NEXUS 870 Fourier-transform infrared (FTIR) spectrometer was employed to obtain the FTIR spectra. Samples were mixed with KBr powder and pressed into pellets for the measurements with a wave number range set at 4000–500 cm$^{-1}$ and a resolution of 1 cm$^{-1}$. A total of 64 scans were accumulated to reduce the spectra noise. Magnetization measurements were obtained at room temperature using a vibrating sample magnetometer (VSM, 8600 Series Magnetometer) with a maximum magnetic field of 16 kOe and a step size of 1 kOe.

## Cell viability

Fibroblast cells (NIH/3T3) were cultured in a 37 °C, 5% CO$_2$ humidified incubator. Dulbecco's modified Eagle's medium (DMEM), supplemented with FBS (10% (v/v)) and penicillin/streptomycin (1% (v/v)), was used as the culture medium. The cytotoxicity of the hydrogels was evaluated using a Live/Dead assay kit (Invitrogen). Circular films of various hydrogels were prepared and washed with 1 wt% NaCl solution, DI water, and ethanol for 7 days to remove the unreacted materials. The prepared films were also sterilized via UV exposure for 2 h. Cells were then cultured in 6-well plates at an initial density of $5 \times 10^4$ cells per well and incubated with sterilized films for 5 days. Cells were stained with Calcein AM (0.5 μL/mL in PBS) and ethidium homodimer-1 (2 μL/mL in PBS) for visualizing live and dead cells, respectively, on days one, three, and five, according to the manufacturer's protocol. Stained samples were imaged using a Zeiss LSM 700 confocal microscope (Carl Zeiss AG, Germany). Image analysis was then performed using ImageJ software. To determine cell viability, the number of green (living) cells was divided by the total number of cells in each image.

## Reporting summary

Further information on research design is available in the Nature Portfolio Reporting Summary linked to this article.

# Data availability

The data that support the findings of this study are presented in the main article and the Supplementary Materials. The raw data are available from the corresponding author upon request.

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

## Acknowledgements

This work was supported by the Natural Sciences and Engineering Research Council of Canada (NSERC). Authors thank Dr. Mahla Poudineh and Natalie Pinchin for their assistance in freeze drying and VSM measurements, respectively.

## Author contributions

H.S. and R.N. conceived the idea and designed the study. R.N., N.B., J.H., and S.J. synthesized hydrogels. R.N., N.B., J.H., and S.J. and T.M. performed characterizations. R.N. and N.B. carried out the robotic demos. H.G. and X.T. performed the cell viability study. A.A. performed the finite element simulations. R.N. and H.S. analyzed and interpreted the results. R.N., N.B., and H.S. wrote the manuscript. H.S., T.M., and X.T. edited the manuscript. H.S. supervised the research.

## Competing interests

The authors declare no competing interests.
