## [Peer Review File · Nature Communications]

Programmable Nanocomposites of Cellulose Nanocrystals and Zwitterionic Hydrogels for Soft RoboticsEditorial Note: Parts of this Peer Review File have been redacted as indicated to remove third-party material where no permission to publish could be obtained.

REVIEWER COMMENTS

Reviewer #1 (Remarks to the Author):

This manuscript concerns the preparation of pH responsive hydrogels prepared from zwitterionic polymers and cellulose nanocrystals for soft robotics applications. Overall, the manuscript is well-written with appropriate citations, and is of significant interest to a broad scientific audience. However, several key points should be addressed prior to further consideration for publication:

1. Several statements/conclusions throughout the manuscript could be better supported via the addition of appropriate supplemental experiments/control conditions.

a) Page 7: "However, the swelling/deswelling of GelWC samples with changes in the pH was not repeatable and the hydrogel started to degrade after 3 consecutive times of exposure to high and low pHs [...] we found that hydrogels with 167:1 weight ratio of comonomers:BIS (molar ratio 143:1) show reversible swelling/deswelling". This type of cyclic swelling/deswelling should be shown for several of the prepared samples (GelWC, Gel, iGel, aGel) to support these claims.

b) Page 9: "The results in Fig. S3, Supplementary Information, show that CNC nanoparticles reorient and align parallel to the flow direction". This is incorrect - the results simply show shear-thinning behavior. Having data for Gel samples as well could help distinguish between shear-thinning in the hydrogel and alignment of CNCs. However, to definitively characterize CNC alignment, techniques such as WAXS/SAXS would be required.

c) Page 11: the authors describe effects of varying gap height on the gradient of microstructural anisotropy, yet this data is not shown.

d) Page 11: the authors also mention that by increasing the pH or ionic strength, the

anisotropic swelling becomes more pronounced. This data is not shown, and effects of ionic strength are not considered throughout the manuscript.

e) The authors mention that due to increased swelling, the AGel samples (with CNC) show decreased shear strength compared to Gel samples (Fig 4a). Yet this trend is reversed for tensile measurements (Fig 3e). A longer discussion surrounding this point would be beneficial. In addition, showing shear strength data for iGel samples would also be beneficial.

f) The swelling data shown in Fig 3d does not match the trends in swelling data shown in the SI, Fig S6. In the SI, iGel and AGel samples swell the most, with Gel samples swelling the least. In Fig 3d, the Gel samples (in both directions) swell significantly more than the iGel samples (in both direction) and the AGel sample in one direction.

g) Having SEM images for iGel samples would be beneficial to compare to those for AGel samples (Fig 3c).

h) Since the authors tested the anisotropic swelling for Gel and iGel samples (Fig 3d), this data could also be shown for tensile testing (Fig 3e).

i) G'' data is missing in Fig 4a, yet is discussed in the manuscript on page 14. The authors also mention that G' dropped to below G'' upon increasing strain (Fig 4b,c); this is not the case for the 2nd and 3rd cycle in Fig 4c. Having similar data for iGel would also be beneficial here.

j) It would be beneficial to mention the time-scale for the various actuation experiments (this is shown in Fig S7 and can be inferred from several of the Supporting Videos, but should nevertheless be discussed).

k) With regards to cell viability testing of the hydrogels, the authors make the following statement on page 19 "The hydrogel prepared with the filliped weight ratio of comonomers showed high toxicity leading to the death of most of the cells on day one of the experiments confirming the effect of zwitterionic monomer on the biocompatibility of the hydrogel". This

data is not shown. Moreover, the authors mention greater than 95% cell viability, yet it is not mentioned how this is calculated. It would also be beneficial to include data for the Gel and iGel/aGel hydrogels in 10 wt% NaCl solution to compare to the GelWC data.

Furthermore, in a practical sense related to the biomedical applicability of such materials, if actuation occurs by changing the pH from 3 to 12, how applicable is this system in biomedicine, where straying from physiologic pH can be exceedingly harmful.

l) For the untethered robotic applications, the authors discuss how the hydrogel can grab a cargo, yet this aspect of grabbing/twisting around the object is not demonstrated.

2. Mention of number of repeats/error is missing for several experiments (mechanical testing, self-healing efficiency/tensile strength)

3. Several aspects of the experimental methodology should be elaborated upon/clarified.

For example:

a) for the self healing experiments, it appears as if hydrogel segments were not joined end-on-end, but rather overlapping. This would create an area with thicker cross-section, which of course should not fracture before an area with thinner cross-section.

b) the shearing mechanism should be explained in more detail. How was repeatability ensured via manual shearing of hydrogels suspensions?

c) For TEM measurements, it is stated that samples are prepared from a powder, yet it is mentioned that MCNC are stored in ethanol. No drying procedure is discussed.

d) Swelling measurements are not discussed at all.

Additionally, several minor points could also be considered to further improve the quality of the manuscript:

1. Figure labels, especially concerning schematics, are often not descriptive enough. e.g. In

Fig 1. e.ii) the square substrate and use of magnets are not explained.

2. In Fig 1., the description mentions a TEM of CNCs, yet the figure implies the TEM image is of nanocomposite hydrogels with shear-aligned CNCs.

3. In Fig S1.b) the labels are incorrect - the text states the hydrogel was first exposed to pH 3 and then pH 12. The label states the opposite.

4. In Fig S2. it is curious that the FTIR peaks at $\sim 2900\text{cm}^{-1}$ for aGel and iGel are different, yet the nanocomposites are (should be) chemically identical.

5. On Page 11, the SI figure numbers are incorrect.

6. The ii and iii labels in Fig 5a are incorrect. In addition, the blue-to-red gradient is not defined for this Figure.

Reviewer #2 (Remarks to the Author):

The authors report work on the development of cellulose nanocrystal reinforced zwitterionic polymeric hydrogels for application within soft robotics. The zwitterionic functionality is motivated in a biocompatibility context where the material class is known for its anti-fouling properties. A number of different known structuring and alignment concepts are used for controlled actuation and movement claimed to be induced by changing the local surrounding pH of the hydrogels. The work is detailed and comprehensive however some general novelty and depth on the mechanistic characterizations are missing. Specific points of feedback are included below.

P2: The conceptualization and realistic potential use cases that need zwitterionic / anti-fouling properties should be reinforced. At the moment this functionality looks like more of an add on although it is a key property of the presented material concept.

P3: The terminology “programmed” suggest capabilities beyond simple stimuli-

responsiveness and material / robotics induced properties that can be activated in more than one single step. To which extent can the presented hydrogels really be “programmed”?

P5: Data on the actual alignment of the CNCs appear to be missing and should be included. The presented SEM suggest actually relatively poor and closer to random organization.

P5: Motivation of structuring patterns, please explain why the specific and for instance helical motif was chosen for the grippers?

P6-7: A detailed description and understanding of the swelling mechanism is missing and critically needs to be included. It is not clear how a structure having both anionic and cationic charges, which I assume both are pH dependent, can swell as a result of repelling anionic charge at high pH and contract at low pH where we would still expect to have similarly repelling cationic charges? The discussion and more importantly analysis of this critical functionality needs to be included.

P7: Linked to the above question, why was this specific comonomer to BIS ratio chosen? Why were other formulation and ratios not giving reversible actuation behavior?

P9: Isotropic properties are claimed for casted and non-sheared samples. However, substrate interaction can induce alignment. Data should be included verifying the isotropic nature of these samples.

P12-13: It is claimed that CNCs is a major source of birefringence in the samples and that stretching can cause complete rearrangements of the CNCs affecting the optical properties of the samples. It should be clarified using control samples not containing CNCs which optical effects could potentially be ascribed to the presence of CNCs. Also, even though the strains are large it is difficult to imagine a full 90 degree rotation of CNCs and these orientation effects should be supported by additional quantitative experiments for instance using x-ray scattering.

P16: The bimorphic structure of these samples is one of the more interesting findings of the

paper since this allows for the creation of structured samples within one single material. In line with previous comments, this structural / alignment gradient across the sample thickness should be characterized in more detail. Also the FEM modeling should be used to predict more complex shape movements as the conclusions that these simulations are now supporting are such that could have been foreseen without any simulation based purely on intuition and in fact several previously published works. What other more precise information other than the main bending deformation could be the FEM simulations give you?

P21: For the practical application of these materials for soft robotics, how do you imagine the pH switch being used? Which are the realistic use cases and how would you be able to non-invasively trigger this actuation?

P23: Sustainability is introduced both in the introduction and in the conclusion of this paper. Even though CNCs, which arguably could be claimed to be sustainable, the use of other synthetic materials and processing does not necessarily make this into a sustainable hydrogel composite. Without a proper sustainability analysis and motivation I believe that the concept of sustainability is a stretch and should not be used here.

Reviewer #3 (Remarks to the Author):

This paper presented the responsive hydrogel nanocomposites largely composed of zwitterionic monomers and geometrically asymmetric CNC particles, showing predetermined microstructural anisotropy, shape-transformation, self-healing behavior, cytocompatibility, and acceptable mechanical properties. Meanwhile, this paper also demonstrates a tethered gripper and an untethered spiral robot capable of soft and light cargo transport.

Overall, I think this paper contains some interesting experimental results. However, the work has several deficiencies (explained in detail below) that make it impossible for me to recommend publication of the article in such a high-level journal like Nature Communications in the present form. Detailed comments are as follow:

1)In Figures 1(d) and (e), the schematic diagrams of sheared anisotropic hydrogels show

inconsistent arrangement densities of CNC particles. Although these are conceptual illustrations, it is essential to ensure reasonable and realistic representation.

2) In the section of physically-crosslinked hydrogels, the authors claim that physically crosslinked hydrogel with a 3:1 DMAPS:MAA weight ratio (called GelWC hereafter), although soft, is still mechanically robust and practical for soft robotic applications. I don't understand why this judgment can be made, and I hope more explanations can be given. What literature or experiments can prove that the performance of this material ratio can be used in the application of soft robots.

3) P(DMAPS-MAA) copolymers are proven sensitive to temperature. Are this hydrogel nanocomposites (self-healing ability, deformation speed, longevity, durability, etc.) also sensitive to temperature? As a potential candidate for biomedical robotic applications, this material needs to adapt to the temperature inside the organism and be verified experimentally in similar simulation environment.

4) In the section of soft robotic application, the authors demonstrate a pH-responsive tethered 4-finger soft gripper for soft and light cargo transport. Although it was a conceptual demonstration, readers might be interested in certain characterization properties of the actuator. For example, the deformation response speed, sensitivity, and gripping strength of the gripper after being exposed to acid or alkaline solutions. The supplementary videos show that the grasping experiment is carried out in a solution, and the buoyancy force of the grasped object is obviously greater than the gravity of itself. Can this kind of soft gripper application find similar practical application scenarios?

5) In this paper, the proposed materials and conceptual demonstrations in the field of soft robotics have significant potential applications in medicine, such as drug delivery within biological organisms. However, many biological organs, such as the gastrointestinal tract, often exhibit an acidic environment for sterilization purposes and do not have an absolute neutral pH value. This raises the question of whether it would affect the deformation behavior of the material. Additionally, once the robot reaches a specific organ site, it becomes important to consider how we can externally deliver acid solution to the robot to trigger the release of the drug. The robotics experiments were all carried out in solution. In actual environments, such as inside the human body, it is very difficult to change the overall pH value (3~12) of the environment. These are indeed important issues that could be mentioned in the future outlook section.

6) Is it possible to achieve a specific deformation by locally dripping an acidic/alkaline solution on the nanocomposites in air environment?

7) The authors describe the self-healing process of the material as follows in the article:

“First, we studied the effect of time on the self-healing of GelWC and noticed a gradual enhancement of healing efficiency with time. After about 4 h GelWC showed full recovery, Fig. 2b.” and “The comparison between the results of mechanical testing on original physically crosslinked hydrogels of different formulation and their self-healed counterparts after 6 h also revealed the complete recovery of mechanical properties, or self-healing efficiency of 100%, containing at least 75 wt% of DMAPS (Fig. 2c).” My questions are:

1. During the self-healing process, the expressions about the healing results of 4 hours and 6 hours are basically the same, but we can clearly see the difference in the final healing effect in Fig. 2b. And the healing effect of 6 hours was significantly lower than that of 4 hours, and the author did not analyze the reasons for the decline in the article. It is also not indicated whether the proportion of its materials, environment and other factors are completely consistent during the healing process?

2. At the same time, in the description of “self-healing efficiency of 100%, containing at least 75 wt% of DMAPS (Fig. 2c).”, there is no clear determination of the content of DMAPS, if its content is in the self-healing process. If there is a change, is the healing effect of the material under different raw material ratios accurate? What is the obvious effect of the content of DMAPS on the self-healing effect of the material itself?

Programmable Nanocomposites of Cellulose Nanocrystals and Zwitterionic Hydrogels for Soft Robotics

*Rasool Nasser, Negin Bouzari, Junting Huang, Hossein Golzar, Sarah Jankhani, Xiaowu (Shirley) Tang, Tizazu H. Mekonnen, Amirreza Aghakhani and Hamed Shahsavan**

Response letter to reviewers

NCOMMS-23-10872-T

We are grateful to the reviewers for their detailed comments that helped us improve our manuscript significantly. We have addressed the comments from each reviewer point-by-point in this response letter. For clarity purposes, we have marked our response to the reviewer's comment in blue. We have revised the main text accordingly and highlighted the changes in this letter and the main text in yellow to facilitate the revision. We hope our revisions address the reviewers' questions and suggestions and that the revised manuscript meets the standards for publication in Nature Communication.

REVIEWER COMMENTS

Reviewer #1 (Remarks to the Author):

This manuscript concerns the preparation of pH responsive hydrogels prepared from zwitterionic polymers and cellulose nanocrystals for soft robotics applications. Overall, the manuscript is well-written with appropriate citations, and is of significant interest to a broad scientific audience. However, several key points should be addressed prior to further consideration for publication:

1. Several statements/conclusions throughout the manuscript could be better supported via the addition of appropriate supplemental experiments/control conditions.

a) Page 7: "However, the swelling/deswelling of GelWC samples with changes in the pH was not repeatable and the hydrogel started to degrade after 3 consecutive times of exposure to high and low pHs [...] we found that hydrogels with 167:1 weight ratio of comonomers:BIS (molar ratio

143:1) show reversible swelling/deswelling". This type of cyclic swelling/deswelling should be shown for several of the prepared samples (GelWC, Gel, iGel, aGel) to support these claims.

We included the swelling deswelling data for all the hydrogels (GelWC, Gel, iGel, and aGel) in the supporting information (Fig. S2, Supplementary Information) and changed the manuscript accordingly. For GelWC, Gel, and iGel in which we did not have shape deformation upon swelling, we reported the degree of swelling $(L - L_0)/L_0$ and for aGel we reported the bending angle (defined as the angle between a vertical tangent applied to one edge of the hydrogel and a vector connecting two edges of the hydrogel after bending).

Changes in the manuscript:

Page 7: However, the swelling/deswelling of GelWC samples with changes in the pH was not repeatable and the hydrogel started to degrade after the first cycle (Fig. S2a, Supplementary Information).

Page 7: After a systematic variation of BIS concentration, along with swelling/deswelling experiments, we found that hydrogels with 167:1 weight ratio of comonomers:BIS (molar ratio 143:1) show reversible swelling/deswelling, which are called Gel hereafter (Fig. S2b, Supplementary Information).

Page 19: Fig. S2d shows the reversible bending and unbending of a short strip of aGel for 7 cycles in response to pH.

b) Page 9: "The results in Fig. S3, Supplementary Information, show that CNC nanoparticles reorient and align parallel to the flow direction". This is incorrect - the results simply show shear-thinning behavior. Having data for Gel samples as well could help distinguish between shear-thinning in the hydrogel and alignment of CNCs. However, to definitively characterize CNC alignment, techniques such as WAXS/SAXS would be required.

We agree with the reviewer's point that the shear-thinning behavior of the aGel precursor alone cannot confirm the reorientation and alignment of CNCs. We revised the statement in the manuscript. However, we believe that it indicates the possibility of the reorientation and alignment of CNCs. That is the reason we have done a range of characterization including imaging between crossed-polarizers, POM, SEM, 2D-SAXS, and 2D-XRD to confirm the alignment of CNCs in the hydrogel. We included the viscosity vs shear rate data for the precursor of the Gel sample (Fig. S4a, Supplementary Information). Newtonian behavior with no shear-thinning of this precursor confirms the role of CNC and its orientation in the shear-thinning behavior of the iGel and aGel precursor. We also added the images of Gel between crossed-polarizers (Fig 3ai, 3aai) and its POM images at different angles (Fig. S4). No birefringence is observed for Gel indicating that the source of birefringence is CNC nanoparticles. The orientation of CNCs in hydrogels was also evaluated by 2D-XRD. For Gel, the diffraction pattern was completely circular and angle independent indicating the amorphous

structure of the material (Fig. 3bi). A rather weak and slightly angle-dependent diffraction pattern at all azimuthal angles on the diffraction at $2\theta = 22.9^\circ$ (corresponding to the (200) diffraction of cellulose $I\beta$ crystal in CNCs) was observed for IGel which suggests that domains of parallel CNCs exist within the film but randomly oriented (polydomain) (Fig. 3bii). The diffraction pattern of AGel, however, showed equatorial arcs and strong angle dependence, revealing the unidirectional alignment of CNCs along the shearing direction (monodomain) (Fig. 3biii). The intensity with respect to the azimuthal angle, $I(\phi)$, on the (200) scattering plane also confirmed the alignment of CNCs along the shearing direction in AGel (Fig. 3f). To quantify the extent of alignment, The Hermans order parameter (S) was calculated based on the diffraction intensities at $2\theta = 22.9^\circ$ (details presented in Supplementary Information). The values of S are 0.03, 0.49, and 0.74 for Gel, IGel, and AGel, respectively, indicating the isotropic structure of Gel and a higher degree of anisotropy in the plane of the film for AGel compared to IGel. We also included the small angle x-ray scattering (SAXS) results of AGel, IGel, and Gel. For Gel the diffraction pattern was completely circular and angle independent. The diffraction pattern of IGel was oval-shaped and slightly angle-dependent and for AGel it was completely elongated which further confirmed the 2D-XRD results (Fig. 3ci, ii, and iii).

Changes in the manuscript:

Page 9: The results in Fig. S4, Supplementary Information, show that CNC nanoparticles can reorient and align upon application of shear. No shear thinning was observed for Gel precursor without CNC which confirms the role of CNC and its orientation in the shear thinning behavior of the precursor.

Page 10: No interference colors were observed in the images of Gel taken between crossed polarizers or with a POM that confirms the role of CNCs in the interference colors (Fig. 3ai and ii, and Fig. S5ai and ii, Supplementary Information).

Page 10: The orientation of CNCs in hydrogels was also evaluated by two-dimensional X-ray diffraction (2D-XRD). For Gel the diffraction pattern was completely circular and angle independent indicating the amorphous structure of the material (Fig. 3bi). A rather weak and slightly angle-dependent diffraction pattern at all azimuthal angles on the diffraction at $2\theta = 22.9^\circ$ (corresponding to the (200) diffraction of cellulose $I\beta$ crystal in CNCs) was observed for IGel which suggests that CNCs are parallel in the film but randomly oriented in all directions (polydomain) (Fig. 3bii). The diffraction pattern of AGel, however, showed equatorial arcs and strong angle dependence, revealing the unidirectional alignment of CNCs along the shearing direction (monodomain) (Fig. 3biii). The intensity with respect to the azimuthal angle, $I(\phi)$, on the (200) scattering plane also confirmed the alignment of CNCs along the shearing direction in AGel (Fig. 3f). To quantify the extent of alignment, The Hermans order parameter (S) was calculated based on the diffraction intensities at $2\theta = 22.9^\circ$ (details presented in Supplementary Information). The values of S are 0.03, 0.49, and 0.74 for Gel, IGel, and AGel, respectively, suggesting the isotropic structure of Gel and a higher degree of anisotropy in the plane of the

film for AGel compared to IGel. We also performed two-dimensional small-angle x-ray scattering (2D-SAXS) on AGel, IGel, and Gel. For Gel the diffraction pattern was completely circular and angle independent. The diffraction pattern of IGel was oval-shaped and slightly angle-dependent and for AGel it was completely elongated which further confirmed the unidirectional alignment of CNCs along the shearing direction (Fig. 3ci, ii, and iii). Fig. S5d shows the 1D radial-averaged SAXS plots of Gel, IGel, and AGel. The low-q curves show a $\sim q^{-1}$ asymptote, which is because the length of CNC is much larger than its width. At large-q values, all profiles exhibit a $\sim q^{-4}$ asymptote, which can be explained by the presence of a sharp interface.

c) Page 11: the authors describe effects of varying gap height on the gradient of microstructural anisotropy, yet this data is not shown.

We added the SEM image of the cross-section of AGel with 250 μm thickness to the Supplementary Information (Fig. 6).

Changes in the manuscript:

Page 12: The gradient of microstructural anisotropy was less pronounced for films of 250 μm thickness (Fig. S6, Supplementary Information).

d) Page 11: the authors also mention that by increasing the pH or ionic strength, the anisotropic swelling becomes more pronounced. This data is not shown, and effects of ionic strength are not considered throughout the manuscript.

We added the pictures of shape changes of AGel pieces cut with the CNC alignment perpendicular to the long axis, parallel to the long axis, and 45° angle with respect to the long axis in response to ionic strength in a 2M NaCl solution to the Supplementary Information (Fig. S10).

Changes in the manuscript:

Page 19: It is also possible to trigger the shape change in AGel strips by swelling them in a saline solution. Fig. S10 shows the pictures of the shape changes of AGel pieces in response to ionic strength in a 2M NaCl solution after 5 mins.

e) The authors mention that due to increased swelling, the AGel samples (with CNC) show decreased shear strength compared to Gel samples (Fig 4a). Yet this trend is reversed for tensile measurements (Fig 3e). A longer discussion surrounding this point would be beneficial. In addition, showing shear strength data for iGel samples would also be beneficial.

As depicted in Fig. S7 Gel has the highest storage modulus followed by IGel and then AGel. We attributed this behavior to the reverse trend in the equilibrium swelling ratio of the hydrogels

(Fig. S9). In the linear viscoelastic range for which the storage modulus of Gel is higher than AGel, the deformations are minuscule so short-range electrostatic interactions as the source of physical crosslinking can stay intact and the plasticization of water molecules is the main reason for the difference between mechanical properties of different samples. In tensile tests, however, the deformations are large so electrostatic interactions and physical crosslinks are disrupted. As a result, chemical crosslinks and CNCs as nano-reinforcers play a more significant role. That is the reason for the higher tensile modulus of hydrogels containing CNC compared to Gel. Frequency sweep data for iGel is presented in Fig. S7a. We added this discussion to the manuscript.

Changes in the manuscript:

Page 13: In the linear viscoelastic range for which the storage modulus of Gel is higher than AGel, the deformations are minuscule so short-range electrostatic interactions as the source of physical crosslinking can stay intact and the plasticization of water molecules is the main reason for the difference between mechanical properties of different samples. In tensile tests, however, the deformations are large so electrostatic interactions are disrupted. As a result, CNCs as nano-reinforcers play a more significant role. That is the reason for the higher tensile modulus of hydrogels containing CNC compared to Gel.

f) The swelling data shown in Fig 3d does not match the trends in swelling data shown in the SI, Fig S6. In the SI, iGel and AGel samples swell the most, with Gel samples swelling the least. In Fig 3d, the Gel samples (in both directions) swell significantly more than the iGel samples (in both directions) and the AGel sample in one direction.

The swelling data presented in Fig. 3d show the change in the length of a rectangle cut from different hydrogels. We put a rectangular piece of the hydrogel into the water and recorded the sample from the top to extract this data from the frames of the recording at different times. The data presented in Fig. S9, however, is collected by weighing the samples at different times and calculating the weight increase. Because we did not record the thickness of the hydrogel samples during the experiment, these two data sets are not comparable.

Changes in the manuscript:

Supplementary Information, Page 6: The data presented in this figure is collected by weighing the samples at different times and calculating the weight increase. Because we did not record the thickness of the hydrogel samples during the experiment, this data is not comparable with the data presented in Fig. 3g.

g) Having SEM images for iGel samples would be beneficial to compare to those for AGel samples (Fig 3c).

We presented the SEM image of the cross-section of iGel in Fig. S6b. This image revealed layered structures with periodic spacing and spiral stacking of CNCs that are characteristic of

polydomain microstructure. We added the discussion on the difference between SEM of AGel and IGel to the manuscript.

Changes in the manuscript:

Page 12: To compare the microstructure, the SEM image of the cross-section of IGel presented in Fig. S6b This image revealed layered structures with periodic spacing and spiral stacking of CNCs that are characteristic of chiral nematic assemblies.

h) Since the authors tested the anisotropic swelling for Gel and iGel samples (Fig 3d), this data could also be shown for tensile testing (Fig 3e).

We did at least 3 replicates for all tensile experiments. For Gel and IGel samples whose long axis were either parallel or perpendicular to the long axis of the hydrogels. We added the replicates to the Supplementary Information (Fig. S8) with the labels that show the direction.

Changes in the manuscript:

Page 13: As anticipated, Gel specimens showed almost identical mechanical properties measured in different directions (Fig. S8 Supplementary Information). For IGel, the tensile modulus and tensile strength were slightly higher in the parallel direction compared to the perpendicular direction which confirms minor induced anisotropy during the casting (Fig. S8 Supplementary Information).

i) G'' data is missing in Fig 4a, yet is discussed in the manuscript on page 14. The authors also mention that G' dropped to below G'' upon increasing strain (Fig 4b,c); this is not the case for the 2nd and 3rd cycle in Fig 4c. Having similar data for iGel would also be beneficial here.

We included the G'' for all the samples in Fig. S7b of Supplementary Information. We revised our statement about G' and G'' . Both G' and G'' will decrease upon increasing the strain and recover their original values after removing the high strain and resting. We also included the step-strain data for IGel in Fig. S7c of Supplementary Information. Alignment of CNC seems to help preserve the solid-like properties of the AGel under high strains.

Changes in the manuscript:

Page 15: When Gel and AGel were subjected to a small strain in the linear viscoelastic region (1%, Fig. 4a) for 200 s, G' was greater than G'' . By increasing the strain to a value greater than the linear viscoelastic threshold (100%, Fig. 4a) for 200 s, both G' and G'' dropped immediately indicating the network disruption. After 200 s rest, the strain was reduced to 1% again, hydrogel exhibited complete recovery of both G' and G'' . Alignment of CNC seems to help preserve the solid-like properties of the AGel under high strains in the second and third cycles probably due to the higher available surface area as a result of fewer contacts and more effective interactions

with the matrix. iGel showed similar behavior in the step-strain experiment (Fig. S7c, Supplementary Information).

j) It would be beneficial to mention the time-scale for the various actuation experiments (this is shown in Fig S7 and can be inferred from several of the Supporting Videos, but should nevertheless be discussed.

We added some discussions on the timescale of shape changes. We also added the actual times of different frames of robotic functions to Fig.7.

Changes in the manuscript:

Page 19: The time scale of both shape deformation and recovery was around 5 minutes.

k) With regards to cell viability testing of the hydrogels, the authors make the following statement on page 19 "The hydrogel prepared with the flipped weight ratio of comonomers showed high toxicity leading to the death of most of the cells on day one of the experiments confirming the effect of zwitterionic monomer on the biocompatibility of the hydrogel". This data is not shown. Moreover, the authors mention greater than 95% cell viability, yet it is not mentioned how this is calculated. It would also be beneficial to include data for the Gel and iGel/aGel hydrogels in 10 wt% NaCl solution to compare to the GelWC data. Furthermore, in a practical sense related to the biomedical applicability of such materials, if actuation occurs by changing the pH from 3 to 12, how applicable is this system in biomedicine, where straying from physiologic pH can be exceedingly harmful.

We added the following statement to the Materials and Methods section: "Image analysis was then performed using ImageJ software. To determine cell viability, the number of green (living) cells was divided by the total number of cells in each image."

Also, we added the confocal image of the hydrogel prepared with the flipped weight ratio of comonomers to the supplementary information. Notably, due to the hydrogel's toxicity, most of the cells were detached from the culture tissue plate within the first 24 hrs. Just a few of them remained, which were all dead.

We added pictures of the interaction of all hydrogels with 10 wt% NaCl solution to the manuscript (Fig. S11b). The pKa of methacrylic acid (the pH-responsive component of our hydrogel) is around 4.7. By increasing pH above this level, the actuation will happen. In this manuscript for proof of principle, we have used extreme pHs to make the shape changes faster, however, the hydrogel can undergo actuation even by milder pHs that are in the range of biological pHs (for example pH of the stomach cavity is around 2 and pH of the colon is around 8-8.5).

Changes in the manuscript:

Page 29: Image analysis was then performed using ImageJ software. To determine cell viability, the number of green (living) cells was divided by the total number of cells in each image.

Page 21: The hydrogel prepared with the flipped weight ratio of comonomers showed high toxicity leading to the death and detachment of most of the cells on day one of the experiments confirming the effect of zwitterionic monomer on the biocompatibility of the hydrogel (Fig. S11, Supplementary Information).

1) For the untethered robotic applications, the authors discuss how the hydrogel can grab a cargo, yet this aspect of grabbing/twisting around the object is not demonstrated.

We added the grabbing stage of the robot around the cargo to the video (Movie S12). Note that the cargo-grabbing process is not fully touchless and we had to direct the cargo to the right spot manually. Entirely touchless grasping step requires active control over the positioning of the twisting ribbon, which is not the scope of this work. However, our research in the near future will address the active control of such smart constructs using external magnetic fields. We have added the following description to the main text.

Changes in the manuscript:

Page 25, the caption of Figure 7: Note that the cargo-grabbing process is not fully touchless and we had to direct the cargo to the right spot manually. Entirely touchless grasping step requires active control over the positioning of the twisting ribbon, which is not the scope of this work.

2. Mention of number of repeats/error is missing for several experiments (mechanical testing, self-healing efficiency/tensile strength).

We added the number of replicates and error bars wherever we had replicates. We added the graphs of tensile strength and elongation at break of physically-crosslinked DMAPS-MAA hydrogels containing different ratios of comonomers with the error bars to Fig. S1, Supplementary Information. The average value of at least three replicates is reported. The error bars represent the standard error. We added an error bar to Fig. 2c. The average value of at least three replicates is reported. The error bars represent the standard error. We added the replicates of tensile graphs in Fig. 3e to the Fig. S8, Supplementary Information. Healing efficiencies associated with Fig. 4d and e were calculated using 3 replicates and the average value was reported in the text.

Changes in the manuscript:

Page 8: Caption of Fig. 2.

Page 15: Caption of Fig. 3.

Page 17: Caption of Fig. 4.

3. Several aspects of the experimental methodology should be elaborated upon/clarified. For example:

a) for the self healing experiments, it appears as if hydrogel segments were not joined end-on-end, but rather overlapping. This would create an area with thicker cross-section, which of course should not fracture before an area with thinner cross-section.

We added more details to the methodology of self-healing experiments. We agree with the reviewer about the most probable location of fracture during the tensile test, which will NOT be on the thicker overlapped and healed area. That said, the most probable mode of failure one would expect to see after self-healing of different hydrogel pieces is delamination and detachment of two hydrogel layers in the overlapped region rather than fracture from the bulk.

Changes in the manuscript:

Page 28: To evaluate the self-healing efficiency of hydrogels a strip with dimensions of 1cm × 3cm was cut. The sample was then cut from the middle and two halves were overlapped 5 mm and slight pressure was applied to ensure full contact. After 6 hr tensile experiment was performed on the healed sample.

b) the shearing mechanism should be explained in more detail. How was repeatability ensured via manual shearing of hydrogels suspensions?

We added more details on the shearing mechanism to the manuscript. To make the process more repeatable, aligning was done by a single individual for different samples who was trying to be as consistent as possible. However, this process is not completely reproducible and that is the reason we have a plan to adopt a more repeatable method such as 3D microextrusion printing for making the samples in the future.

Changes in the manuscript:

Page 27: The precursor was transferred on a glass slide between two strips of 500 μm spacer glued to the edges. The precursor was then sheared by the edge of another glass slide at a rate of 5 cm s^{-1} a few times.

c) For TEM measurements, it is stated that samples are prepared from a powder, yet it is mentioned that MCNC are stored in ethanol. No drying procedure is discussed.

We included the drying procedure before making the TEM samples from powder.

Changes in the manuscript:

Page 28: After removing ethanol and drying the MCNCs completely, the powder sample was dispersed in water (1 mg ml^{-1}), and about 10 μl of the dilute dispersion was cast onto the surface of a 300 mesh grid which was then dried at an ambient temperature for 24 hr before taking the images on a Philips CM10 transmission electron microscope.

d) Swelling measurements are not discussed at all.

The swelling experiment was performed in two different ways. The first method was by recording the samples while they are swelling in water and later measuring the dimensions from the video frames and reporting $\Delta L/L_0$. The second method was measuring the weight of samples after swelling at different times in water and reporting $\Delta m/m_0$. The result of the first method was used to show the differential swelling in different directions that lead to the shape deformations and discussed in detail in the manuscript (page 17). The result of the second method was used to describe the rheological results of the hydrogels. Generally, hydrogels containing CNCs (AGel and IGel) were swelling more due to the hydrophilicity of CNCs and their ability to absorb and maintain water. By aligning the CNCs in AGel their ability to absorb water was enhanced probably due to fewer contacts among them which resulted in a higher swelling ratio of AGel compared to IGel. We added this discussion to the Supplementary Information.

Changes in the manuscript:

Supplementary Information, Page 6: Generally, hydrogels containing CNCs (AGel and IGel) were swelling more due to the hydrophilicity of CNCs and their ability to absorb and maintain water. By aligning the CNCs in AGel their ability to absorb water was enhanced probably due to fewer contacts among them and more available surface area which resulted in a higher swelling ratio of AGel compared to IGel.

Additionally, several minor points could also be considered to further improve the quality of the manuscript:

1. Figure labels, especially concerning schematics, are often not descriptive enough. e.g. In Fig 1. e.ii) the square substrate and use of magnets are not explained.

We added more details to the figure captions.

Changes in the manuscript:

Fig.1. caption, Page 5: By adding a magnetic patch to the untethered soft gripper, it can be navigated and steered by an external magnetic field.

Fig. 2. caption, Page 8: At high pH values (greater than 4.7), the -COOH groups of MAA are ionized, and the charged -COO^- groups repel each other, leading to the swelling of the hydrogel while at lower pHs this process is reversed.

Fig. 5. caption, Page 20: Our system can be deemed as a bimorph with nonidentical swelling behavior along the thickness in which CNCs are oriented randomly on one side, with a characteristic thickness of h_I ($\sim 200 \mu\text{m}$), and unidirectionally aligned on the other side, with a characteristic thickness of h_A ($\sim 600 \mu\text{m}$).

Fig. 7. caption, Page 25: By adding a magnetic patch to the untethered soft gripper, it can be navigated and steered by an external magnetic field.

2. In Fig 1., the description mentions a TEM of CNCs, yet the figure implies the TEM image is of nanocomposite hydrogels with shear-aligned CNCs.

We made it clearer.

3. In Fig S1.b) the labels are incorrect - the text states the hydrogel was first exposed to pH 3 and then pH 12. The label states the opposite.

We corrected this mistake.

Changes in the manuscript:

Fig. S1. caption, Supplementary Information, Page 2: The degree of swelling of a GelWC sample in water (pH ~ 7) and in buffers with pH 3 and 12.

4. In Fig S2. it is curious that the FTIR peaks at $\sim 2900\text{cm}^{-1}$ for aGel and iGel are different, yet the nanocomposites are (should be) chemically identical.

These peaks are characteristics of the C-H stretching of CNC nanoparticles. As highlighted by dashed lines, these peaks are present in both spectra. The only difference between the two

spectra is the sharpness of the peaks in AGel which can be attributed to the alignment of CNCs and fewer contacts and hydrogen bonding among them.

Figure 1. FTIR spectra of AGel and IGel.

5. On Page 11, the SI figure numbers are incorrect.

We corrected this mistake.

6. The ii and iii labels in Fig 5a are incorrect. In addition, the blue-to-red gradient is not defined for this Figure.

We corrected this mistake. Blue-to-red gradients show the total displacements in mm. We added this explanation to the caption of Figure 5.

Changes in the manuscript:

Fig. 5. caption, Page 20: Blue-to-red gradients show the total displacements in mm.

Reviewer #2 (Remarks to the Author):

The authors report work on the development of cellulose nanocrystal reinforced zwitterionic polymeric hydrogels for application within soft robotics. The zwitterionic functionality is motivated in a biocompatibility context where the material class is known for its anti-fouling properties. A number of different known structuring and alignment concepts are used for controlled actuation and movement claimed to be induced by changing the local surrounding pH of the hydrogels. The work is detailed and comprehensive however some general novelty and

depth on the mechanistic characterizations are missing. Specific points of feedback are included below.

P2: The conceptualization and realistic potential use cases that need zwitterionic / anti-fouling properties should be reinforced. At the moment this functionality looks like more of an add on although it is a key property of the presented material concept.

We highlighted the importance of the anti-fouling and biocompatibility requirements of hydrogel-based robots in biomedical applications in the introduction.

Changes in the manuscript:

Page 3: Biomedical hydrogel soft robots are naturally susceptible to being fouled by biomolecules and prone to trigger foreign body reactions (FBR).^{31,32} Aside from the health concerns they cause, biofouling and FBR disrupt the intended functionality of soft robots. Zwitterionic hydrogels are known for their superior anti-fouling and biocompatibility properties.^{33,34} They possess super hydrophilicity, zero net charges, and H-bond accepting functional groups, which leads to minimal protein adsorption and cell adhesion.³² These characteristics make them great candidates for designing miniaturized medical soft robots with minimal FBR. To the best of our knowledge, anti-fouling zwitterionic hydrogels have never been used as stimuli-triggered shape-morphing materials, especially for soft robotic applications.

P3: The terminology “programmed” suggest capabilities beyond simple stimuli-responsiveness and material / robotics induced properties that can be activated in more than one single step. To which extent can the presented hydrogels really be “programmed”?

The stimuli-responsiveness of a hydrogel provides it with only isotropic swelling/deswelling in response to an external stimulus. Although this isotropic swelling/deswelling is important in some applications, shape morphing requires a more sophisticated microstructural assembly that enables differential swelling/deswelling in different directions. Inducing anisotropy to the microstructure of the hydrogel to achieve this differential swelling/deswelling is a crucial part of shape change programming. cut-and-paste strategy owing to the self-healing properties of this hydrogel is another tool for shape change programming of this zwitterionic system. Considering all the steps that should be taken to achieve the final desired shape to enable certain functionality, we believe that using “programming” is not only fair but required to fully describe our approach. Note that in this work, and for the proof of concept, we have only shown the unidirectional shear as a means to program the shape-change of our hydrogels. This technique can be extended, and more sophisticated deformation profiles can be achieved by the use of multi-directional or complex shearing strategies, for instance by using 3D printing.

P5: Data on the actual alignment of the CNCs appear to be missing and should be included. The presented SEM suggest actually relatively poor and closer to random organization.

We believe the shear-thinning behavior of the AGel precursor indicates the possibility of the reorientation and alignment of CNCs. That is the reason we have done a range of characterization including imaging between crossed-polarizers, POM, SEM, 2D-SAXS, and 2D-XRD to confirm the alignment of CNCs in the hydrogel. We included the viscosity vs shear rate data for the precursor of the Gel sample (Fig. S4a, Supplementary Information). Newtonian behavior with no shear-thinning of this precursor confirms the role of CNC and its orientation in the shear-thinning behavior of the IGel and AGel precursor. We also added the images of Gel between crossed-polarizers (Fig 3ai, 3aii) and its POM images at different angles (Fig. S4). No birefringence is observed for Gel indicating that the source of birefringence is CNC nanoparticles. The orientation of CNCs in hydrogels was also evaluated by 2D-XRD. For Gel the diffraction pattern was completely circular and angle independent indicating the amorphous structure of the material (Fig. 3bi). A rather weak and slightly angle-dependent diffraction pattern at all azimuthal angles on the diffraction at $2\theta = 22.9^\circ$ (corresponding to the (200) diffraction of cellulose $I\beta$ crystal in CNCs) was observed for IGel which suggests that CNCs are parallel in the film but randomly oriented in all directions (polydomain) (Fig. 3bii). The diffraction pattern of AGel, however, showed equatorial arcs and strong angle dependence, revealing the unidirectional alignment of CNCs along the shearing direction (monodomain) (Fig. 3biii). The intensity with respect to the azimuthal angle, $I(\phi)$, on the (200) scattering plane also confirmed the alignment of CNCs along the shearing direction in AGel (Fig. 3f). To quantify the extent of alignment, The Hermans order parameter (S) was calculated based on the diffraction intensities at $2\theta = 22.9^\circ$ (details presented in Supplementary Information). The values of S are 0.03, 0.49, and 0.74 for Gel, IGel, and AGel, respectively, indicating the isotropic structure of Gel and a higher degree of anisotropy in the plane of the film for AGel compared to IGel. We also included the small angle x-ray scattering (SAXS) results of AGel, IGel, and Gel. For Gel the diffraction pattern was completely circular and angle independent. The diffraction pattern of IGel was oval-shaped and slightly angle-dependent and for AGel it was completely elongated which further confirmed the 2D-XRD results (Fig. 3ci, ii, and iii).

Changes in the manuscript:

Page 9: The results in Fig. S4, Supplementary Information, show that CNC nanoparticles can reorient and align upon application of shear. No shear thinning was observed for Gel precursor without CNC that confirms the role of CNC and its orientation in the shear thinning behavior of the precursor.

Page 10: No interference colors were observed in the images of Gel taken between crossed polarizers or with a POM that confirms the role of CNCs in the interference colors (Fig. 3ai and ii, and Fig. S5ai and ii, Supplementary Information).

Page 10: The orientation of CNCs in hydrogels was also evaluated by two-dimensional X-ray diffraction (2D-XRD). For Gel the diffraction pattern was completely circular and angle independent indicating the amorphous structure of the material (Fig. 3bi). A rather weak and slightly angle-dependent diffraction pattern at all azimuthal angles on the diffraction at $2\theta = 22.9^\circ$ (corresponding to the (200) diffraction of cellulose $I\beta$ crystal in CNCs) was observed for IGel which suggests that CNCs are parallel in the film but randomly oriented in all directions (polydomain) (Fig. 3bii). The diffraction pattern of AGel, however, showed equatorial arcs and strong angle dependence, revealing the unidirectional alignment of CNCs along the shearing direction (monodomain) (Fig. 3biii). The intensity with respect to the azimuthal angle, $I(\phi)$, on the (200) scattering plane also confirmed the alignment of CNCs along the shearing direction in AGel (Fig. 3f). To quantify the extent of alignment, The Hermans order parameter (S) was calculated based on the diffraction intensities at $2\theta = 22.9^\circ$ (details presented in Supplementary Information). The values of S are 0.03, 0.49, and 0.74 for Gel, IGel, and AGel, respectively, suggesting the isotropic structure of Gel and a higher degree of anisotropy in the plane of the film for AGel compared to IGel. We also performed two-dimensional small angle x-ray scattering (2D-SAXS) on AGel, IGel, and Gel. For Gel the diffraction pattern was completely circular and angle independent. The diffraction pattern of IGel was oval-shaped and slightly angle-dependent and for AGel it was completely elongated which further confirmed unidirectional alignment of CNCs along the shearing direction (Fig. 3ci, ii, and iii). Fig. S5d shows the 1D radial-averaged SAXS plots of Gel, IGel, and AGel. The low- q curves show an $\sim q^{-1}$ asymptote, which is because the length of CNC is much larger than its width. At large- q values, all profiles exhibit an $\sim q^{-4}$ asymptote, which can be explained by the presence of a sharp interface.

P5: Motivation of structuring patterns, please explain why the specific and for instance helical motif was chosen for the grippers?

Differential swelling of the hydrogel provides us with certain capacities for shape morphing. Differential swelling along the thickness provides us with bending deformation. Differential swelling of the aligned layer of the hydrogel enables the twisting to a helical shape. By changing the direction of the long axis of the hydrogel strip with respect to the alignment direction, we can change the pitch of the helix. To showcase the capabilities that differential swelling gives us, we have chosen the shape changes that are depicted in the manuscript (Fig. 5). Our inspiration for the helical gripper was nature. Similar gripping action can be seen in snakes or in climbing plants.

P6-7: A detailed description and understanding of the swelling mechanism is missing and critically needs to be included. It is not clear how a structure having both anionic and cationic charges, which I assume both are pH dependent, can swell as a result of repelling anionic charge at high pH and contract at low pH where we would still expect to have similarly repelling cationic charges? The discussion and more importantly analysis of this critical functionality needs to be included.

DMAPS (zwitterionic monomer used in our work) has permanent positive and negative charges due to the presence of a quaternary ammonium group and a sulfonate group, respectively. None of these groups are pH dependent, and as reported previously in the literature, the swelling capacity of the hydrogel synthesized with just DMAPS does not change at different pHs (Langmuir 2019, 35, 1146–1155). The pH-responsive component of our hydrogel is methacrylic acid (MAA) which has pKa of around 4.7. At high pH values (greater than 4.7), the -COOH groups of MAA are ionized and the charged -COO^- groups repel each other, leading to the swelling of the hydrogel while at lower pHs this process is reversed.

P7: Linked to the above question, why was this specific comonomer to BIS ratio chosen? Why were other formulation and ratios not giving reversible actuation behavior?

We believe we have elaborated on this with sufficient details in the manuscript. To summarize in here, the problem we addressed by adding the BIS as the chemical crosslinker was the dissolution of the sample after swelling at high pHs due to the lack of chemical cross-linking. But we had to be careful with the amount of chemical crosslinker used. The addition of excess amounts of chemical crosslinker limits the movements of chains required for the healing process. We tried different amounts of comonomers to BIS ratio in our preliminary experiments to find the formulation that gives us reversible swelling/deswelling without compromising the self-healing. To confirm the reversibility, we included the swelling deswelling data for all the

hydrogels (GelWC, Gel, IGel, and AGel) in the supporting information (Fig. S2, Supplementary Information) and changed the text accordingly. For GelWC, Gel, and IGel in which we did not have shape deformation upon swelling, we reported the degree of swelling $(L - L_0)/L_0$ and for AGel we reported the bending angle (defined as the angle between a vertical tangent applied to one edge of the hydrogel and a vector connecting two edges of the hydrogel after bending).

Changes in the manuscript:

Page 7: However, the swelling/deswelling of GelWC samples with changes in the pH was not repeatable and the hydrogel started to degrade after the first cycle (Fig. S2a, Supplementary Information).

Page 7: After a systematic variation of BIS concentration, along with swelling/deswelling experiments, we found that hydrogels with 167:1 weight ratio of comonomers:BIS (molar ratio 143:1) show reversible swelling/deswelling, which are called Gel hereafter (Fig. S2b, Supplementary Information).

Page 19: Fig. S 2d shows the reversible bending and unbending of a short strip of AGel for 7 cycles in response to pH.

P9: Isotropic properties are claimed for casted and non-sheared samples. However, substrate interaction can induce alignment. Data should be included verifying the isotropic nature of these samples.

We agree with the reviewer as the self-assembly of CNCs close to the bottom glass substrate can induce alignment and anisotropy. However, the domain size (or thickness) of such self-assembled structures is very small. We have done SEM on the cross-section of the casted sample and as it can be seen the thickness of this layer is very small compared to the rest of the film which has layered structures with periodic spacing and spiral stacking of CNCs that are characteristic of chiral nematic assemblies. As such surface aligned domains alone do not play a tangible effect on the overall anisotropy.

In detail, we see that the surface alignment in this fashion cannot be translated to the bulk of hydrogel precursor (Carbohydrate Polymers 280 (2022) 119005). Perturbations caused by the shear force on the upper substrate, well beyond the thin surface aligned layer where CNC rods form chiral nematic helical domains, cause a random orientation of chiral nematic domains closer to the bottom substrate. As we move away from the bottom substrate towards the regions experiencing larger shear stress, CNCs align themselves well parallel with the shear. We still believe that the gradient along the thickness renders the bending deformation while the alignment parallel to the shear dictates the direction of bending.

[FIGURE REDACTED]

Figure 2. a) SEM images of IGel cross-section with thickness of 500 μm after drying. Scale bar is 100 μm . b) Schematic of CNC arrangement in the polymer network adopted from (Carbohydrate Polymers 280 (2022) 119005).

Changes in the manuscript:

Page 11: The self-assembly of CNCs close to the bottom glass substrate can induce alignment and anisotropy. However, the domain size (or thickness) of such self-assembled structures is very small. We have done SEM on the cross-section of IGel and as it can be seen in Fig. S6b the thickness of this layer is very small compared to the rest of the film which has layered structures with periodic spacing and spiral stacking of CNCs that are characteristic of chiral nematic assemblies. As such surface aligned domains alone do not play a tangible effect on the overall anisotropy.

P12-13: It is claimed that CNCs is a major source of birefringence in the samples and that stretching can cause complete rearrangements of the CNCs affecting the optical properties of the samples. It should be clarified using control samples not containing CNCs which optical effects could potentially be ascribed to the presence of CNCs. Also, even though the strains are large it is difficult to imagine a full 90 degree rotation of CNCs and these orientation effects should be supported by additional quantitative experiments for instance using x-ray scattering.

We addressed the source of birefringence in response to another reviewer's question. For convenience, we copied that response below. About the change of color during the extension of the sample, we should state that our conclusion was based on a comprehensive experimental work on a similar system reported by Kose et al. (Macromolecules 2019, 52, 5317–5324). We think that we have reorientation in our system due to the extension although this reorientation might not be as much as 90 degrees.

We believe shear-thinning behavior of the AGel precursor indicates the possibility of the reorientation and alignment of CNCs. That is the reason we have done a range of characterization including imaging between crossed-polarizers, POM, SEM, 2D-SAXS, and 2D-XRD to confirm the alignment of CNCs in the hydrogel. We included the viscosity vs shear rate data for the precursor of the Gel sample (Fig. S4a, Supplementary Information). Newtonian behavior with no shear-thinning of this precursor confirms the role of CNC and its orientation in the shear-thinning behavior of the IGel and AGel precursor. We also added the images of Gel between crossed-polarizers (Fig 3ai, 3aai) and its POM images at different angles (Fig. S4). No birefringence is observed for Gel indicating that the source of birefringence is CNC nanoparticles. The orientation of CNCs in hydrogels was also evaluated by 2D-XRD. For Gel the

diffraction pattern was completely circular and angle independent indicating the amorphous structure of the material (Fig. 3bi). A rather weak and slightly angle-dependent diffraction pattern at all azimuthal angles on the diffraction at $2\theta = 22.9^\circ$ (corresponding to the (200) diffraction of cellulose $I\beta$ crystal in CNCs) was observed for IGel which suggests that CNCs are parallel in the film but randomly oriented in all directions (polydomain) (Fig. 3bii). The diffraction pattern of AGel, however, showed equatorial arcs and strong angle dependence, revealing the unidirectional alignment of CNCs along the shearing direction (monodomain) (Fig. 3biii). The intensity with respect to the azimuthal angle, $I(\phi)$, on the (200) scattering plane also confirmed the alignment of CNCs along the shearing direction in AGel (Fig. 3f). To quantify the extent of alignment, The Hermans order parameter (S) was calculated based on the diffraction intensities at $2\theta = 22.9^\circ$ (details presented in Supplementary Information). The values of S are 0.03, 0.49, and 0.74 for Gel, IGel, and AGel, respectively, indicating the isotropic structure of Gel and a higher degree of anisotropy in the plane of the film for AGel compared to IGel. We also included the small angle x-ray scattering (SAXS) results of AGel, IGel, and Gel. For Gel the diffraction pattern was completely circular and angle independent. The diffraction pattern of IGel was oval-shaped and slightly angle-dependent and for AGel it was completely elongated which further confirmed the 2D-XRD results (Fig. 3ci, ii, and iii).

Changes in the manuscript:

Page 9: The results in Fig. S4, Supplementary Information, show that CNC nanoparticles can reorient and align upon application of shear. No shear thinning was observed for Gel precursor without CNC that confirms the role of CNC and its orientation in the shear thinning behavior of the precursor.

Page 10: No interference colors were observed in the images of Gel taken between crossed polarizers or with a POM that confirms the role of CNCs in the interference colors (Fig. 3ai and ii, and Fig. S5ai and ii, Supplementary Information).

Page 10: The orientation of CNCs in hydrogels was also evaluated by two-dimensional X-ray diffraction (2D-XRD). For Gel the diffraction pattern was completely circular and angle independent indicating the amorphous structure of the material (Fig. 3bi). A rather weak and slightly angle-dependent diffraction pattern at all azimuthal angles on the diffraction at $2\theta = 22.9^\circ$ (corresponding to the (200) diffraction of cellulose $I\beta$ crystal in CNCs) was observed for IGel which suggests that CNCs are parallel in the film but randomly oriented in all directions (polydomain) (Fig. 3bii). The diffraction pattern of AGel, however, showed equatorial arcs and strong angle dependence, revealing the unidirectional alignment of CNCs along the shearing direction (monodomain) (Fig. 3biii). The intensity with respect to the azimuthal angle, $I(\phi)$, on the (200) scattering plane also confirmed the alignment of CNCs along the shearing direction in AGel (Fig. 3f). To quantify the extent of alignment, The Hermans order parameter (S) was calculated based on the diffraction intensities at $2\theta = 22.9^\circ$ (details presented in Supplementary Information). The values of S are 0.03, 0.49, and 0.74 for Gel, IGel, and AGel, respectively,

suggesting the isotropic structure of Gel and a higher degree of anisotropy in the plane of the film for AGel compared to IGel. We also performed two-dimensional small angle x-ray scattering (2D-SAXS) on AGel, IGel, and Gel. For Gel the diffraction pattern was completely circular and angle independent. The diffraction pattern of IGel was oval-shaped and slightly angle-dependent and for AGel it was completely elongated which further confirmed unidirectional alignment of CNCs along the shearing direction (Fig. 3ci, ii, and iii). Fig. S5d shows the 1D radial-averaged SAXS plots of Gel, IGel, and AGel. The low- q curves show an $\sim q^{-1}$ asymptote, which is because the length of CNC is much larger than its width. At large- q values, all profiles exhibit an $\sim q^{-4}$ asymptote, which can be explained by the presence of a sharp interface.

P16: The bimorphic structure of these samples is one of the more interesting findings of the paper since this allows for the creation of structured samples within one single material. In line with previous comments, this structural / alignment gradient across the sample thickness should be characterized in more detail. Also the FEM modeling should be used to predict more complex shape movements as the conclusions that these simulations are now supporting are such that could have been foreseen without any simulation based purely on intuition and in fact several previously published works. What other more precise information other than the main bending deformation could be the FEM simulations give you?

To study the bimorphic structure we adopted POM and SEM. POM and SEM Images taken from the cross-section of the AGel revealed the presence of two layers along the thickness (Fig. 3dv and 3dvi and e). We also included the POM image of the Gel cross-section (Fig. 3di and 3dii) which confirms the effect of CNCs in birefringence. By comparing the POM image of the cross-section of Gel, IGel, and AGel one can clearly see the bimorphic structure of AGel.

The FEM modeling of mechanical deformations of bimorphic hydrogels is indeed performed using the neo-Hookean hyperelastic model in this manuscript. Our FEM model predicts the swelling behavior of hydrogels and is capable of describing the mechanical behavior of hydrogels in different regimes of large deformation. These large deformations originate from a combination of standard bending and twisting motions, or positive and negative Gaussian curvatures. As shown in (PNAS, July 10, 2018, vol. 115, no. 28, 7206–7211), authors have simulated the most complex shapes, such as a human face, using similar models. More complex deformations can essentially be modeled first by breaking them into local curvatures (Gaussian or other). Likewise, we believe that our model can be applied to more sophisticated deformations from the combination of different types of curvatures with different signs. In fact, our most complex 3D deformation shown in Figure 5f-iii, shows a combination of positive, negative, and zero Gaussian curvatures, which are successfully simulated and verified by the experiments.

P21: For the practical application of these materials for soft robotics, how do you imagine the pH switch being used? Which are the realistic use cases and how would you be able to non-invasively trigger this actuation?

In this manuscript for proof of principle, we have used extreme pHs to make the shape changes faster, however, the actuation can be induced in the hydrogel by milder pHs that are in the range of biological pHs because the pKa of hydrogel should be close to the methacrylic acid that is around 4.7. For instance, we can imagine a soft robot that can be administered to the stomach cavity (pH of around 2). In this low pH soft robot can keep its original shape. By moving the robot to the colon with a pH of around 8-8.5 soft robots can deform to its programmed shape to perform a function. It can then be recovered from the body through the rectum.

P23: Sustainability is introduced both in the introduction and in the conclusion of this paper. Even though CNCs, which arguably could be claimed to be sustainable, the use of other synthetic materials and processing does not necessarily make this into a sustainable hydrogel composite. Without a proper sustainability analysis and motivation I believe that the concept of sustainability is a stretch and should not be used here.

We used the term “sustainability” three times in the manuscript. The first two times were on page 5 as an attribute of CNC sources and when discussing the on-demand degradation of our hydrogel with a specific definition (no waste at the end of its lifetime). The third time was in the conclusion. We believe that the first two occasions are valid but, about the third one, we agree with the reviewer’s point and removed “sustainability”

Changes in the manuscript:

Page 26: The results of this work can expand the development of adaptive and reconfigurable biomimetic soft robots.

Reviewer #3 (Remarks to the Author):

This paper presented the responsive hydrogel nanocomposites largely composed of zwitterionic monomers and geometrically asymmetric CNC particles, showing predetermined microstructural anisotropy, shape-transformation, self-healing behavior, cytocompatibility, and acceptable mechanical properties. Meanwhile, this paper also demonstrates a tethered gripper and an untethered spiral robot capable of soft and light cargo transport.

Overall, I think this paper contains some interesting experimental results. However, the work has several deficiencies (explained in detail below) that make it impossible for me to recommend publication of the article in such a high-level journal like Nature Communications in the present form. Detailed comments are as follow:

1) In Figures 1(d) and (e), the schematic diagrams of sheared anisotropic hydrogels show inconsistent arrangement densities of CNC particles. Although these are conceptual illustrations, it is essential to ensure reasonable and realistic representation.

We revised Fig. 1d and Fig. 5a.

Changes in the manuscript:

Page 5: Fig. 1

Page 20: Fig. 5

2) In the section of physically-crosslinked hydrogels, the authors claim that physically crosslinked hydrogel with a 3:1 DMAPS:MAA weight ratio (called GelWC hereafter), although soft, is still mechanically robust and practical for soft robotic applications. I don't understand why this judgment can be made, and I hope more explanations can be given. What literature or experiments can prove that the performance of this material ratio can be used in the application of soft robots.

For our application, we were looking for a few key properties. First, we wanted to have a soft hydrogel with an elastic modulus in the range of soft body tissues (100 kPa to 1000 kPa (Mater Today (Kidlington). 2011 Mar; 14(3): 96–105.)). The elastic modulus of our hydrogel is around 430 kPa. Second, we wanted a hydrogel with good self-healing properties. Fig. 2c shows that the sample with 3:1 DMAPS:MAA weight ratio is the stiffest sample that shows complete recovery after damage. Finally, we wanted a hydrogel with low cytotoxicity and high biocompatibility. Our cell culture studies (Fig. 6a) showed that the sample with 3:1 DMAPS:MAA weight ratio has a very low cytotoxicity. The sample with the flipped weight ratio (1:3) showed very high cytotoxicity indicating the effect of DMAPS monomer in biocompatibility (Fig. S11). In general, more DMAPS was beneficial in terms of self-healing and biocompatibility and more MAA was beneficial for enhancing the mechanical properties and strength of the hydrogel. The sample with 3:1 DMAPS:MAA weight ratio was the optimum sample considering all the requirements. We added a couple of sentences to elaborate on our rationale.

Changes in the manuscript:

Page 6: Note that the elastic modulus of our hydrogel is in the range of soft body tissues 100 kPa to 1000 kPa, which is essential for non-invasive interaction with as well as manipulation of soft objects, like tissues and cells.

3) P(DMAPS-MAA) copolymers are proven sensitive to temperature. Are this hydrogel nanocomposites (self-healing ability, deformation speed, longevity, durability, etc.) also sensitive to temperature? As a potential candidate for biomedical robotic applications, this material needs

to adapt to the temperature inside the organism and be verified experimentally in similar simulation environment.

We evaluated the shape change of AGel pieces cut with the CNC alignment perpendicular to the long axis and making a 45° angle with the long axis in response to pH change at temperatures close to the physiological temperature. Due to the upper critical solution temperature (UCST) nature of thermo-responsivity of the hydrogel nanocomposite (ACS Nano 2018, 12, 12860–12868), increasing the temperature, increased the speed of shape change (**Fig. S12a** and **Video S13**). The time scale of both shape deformations was around 30 sec. To investigate the effect of physiological temperature on the mechanical properties of the hydrogel, we performed temperature-sweep and strain-sweep tests on AGel. By increasing the temperature, both G' and G'' increased with a slight step at UCST around 57 °C (**Fig. S12b**). The linear viscoelastic region of the hydrogel also slightly increased (**Fig. S12c**). To study the self-healing properties of hydrogel nanocomposite, a step-strain experiment was performed on AGel at 37 °C (**Fig. S12d**). By increasing the strain to a value greater than the linear viscoelastic threshold (100%, **Fig. S12b**) for 200 s, both G' and G'' dropped immediately indicating the network disruption. After 200 s rest, the strain was reduced to 1% again, hydrogel exhibited complete recovery of both G' and G'' .

Changes in the manuscript:

Page 23: Effect of physiological temperature on the hydrogel nanocomposite

We evaluated the shape change of AGel pieces cut with the CNC alignment perpendicular to the long axis and making a 45° angle with the long axis in response to pH change at temperatures close to the physiological temperature. Due to the upper critical solution temperature (UCST) nature of thermo-responsivity of the hydrogel nanocomposite (ACS Nano 2018, 12, 12860–12868), increasing the temperature, increased the speed of shape change (Fig. S12a and Video S13). The time scale of both shape deformations was around 30 sec. To investigate the effect of physiological temperature on the mechanical properties of the hydrogel, we performed temperature-sweep and strain-sweep tests on AGel. By increasing the temperature, both G' and G'' increased with a slight step at UCST around 57 °C (Fig. S12b). The linear viscoelastic region of the hydrogel also slightly increased (Fig. S12c). To study the self-healing properties of hydrogel nanocomposite, a step-strain experiment was performed on AGel at 37 °C (Fig. S12d). By increasing the strain to a value greater than the linear viscoelastic threshold (100%, Fig. S12b) for 200 s, both G' and G'' dropped immediately indicating the network disruption. After 200 s rest, the strain was reduced to 1% again, hydrogel exhibited complete recovery of both G' and G'' .

4) In the section of soft robotic application, the authors demonstrate a pH-responsive tethered 4-finger soft gripper for soft and light cargo transport. Although it was a conceptual demonstration, readers might be interested in certain characterization properties of the actuator. For example, the

deformation response speed, sensitivity, and gripping strength of the gripper after being exposed to acid or alkaline solutions. The supplementary videos show that the grasping experiment is carried out in a solution, and the buoyancy force of the grasped object is obviously greater than the gravity of itself. Can this kind of soft gripper application find similar practical application scenarios?

We added some discussions on the timescale of the deformation response. We also added the actual times of different frames of robotic functions to Fig.7. To evaluate the effect of pH on the mechanical properties of the soft robots, we have done a frequency-sweep on AGel after swelling in a solution with pH of 12 and 3 for about 5 min. Swelling at low pH did not change the dynamic moduli significantly compared to the sample swelled in water. Swelling at high pH, however, increased the dynamic moduli probably due to the stretch of chains as a result of extreme swelling (Fig. S13). As has been mentioned in the question, it is a conceptual demonstration, and for the practical application, we need to have thicker hydrogel films. Due to the limitations of our fabrication technique (inducing alignment with shearing the surface of the hydrogel precursor), we cannot increase the thickness of the sample. In the future, we will consider 3D printing to fabricate thicker hydrogels without losing the CNC alignment.

Changes in the manuscript:

Page 19: The time scale of both shape deformation and recovery was around 5 minutes.

Page 25: Fig.7.

Page 24: To evaluate the effect of pH on the mechanical properties of the soft robots, we have done a frequency-sweep on AGel after swelling in a solution with pH of 12 and 3 for about 5 min. Swelling at low pH did not change the dynamic moduli significantly compared to the sample swelled in water. Swelling at high pH, however, increased the dynamic moduli probably due to the stretch of chains as a result of extreme swelling (Fig. S13).

5) In this paper, the proposed materials and conceptual demonstrations in the field of soft robotics have significant potential applications in medicine, such as drug delivery within biological organisms. However, many biological organs, such as the gastrointestinal tract, often exhibit an acidic environment for sterilization purposes and do not have an absolute neutral pH value. This raises the question of whether it would affect the deformation behavior of the material. Additionally, once the robot reaches a specific organ site, it becomes important to consider how we can externally deliver acid solution to the robot to trigger the release of the drug. The robotics experiments were all carried out in solution. In actual environments, such as inside the human body, it is very difficult to change the overall pH value (3~12) of the environment. These are indeed important issues that could be mentioned in the future outlook section.

In this manuscript for proof of principle, we have used extreme pHs to make the shape changes faster, however, the actuation can be included in the hydrogel by milder pHs that are in the range

of biological pHs because the pKa of hydrogel should be close to the methacrylic acid that is around 4.7. For instance, we can imagine a soft robot that can be sent to the stomach cavity (pH of around 2). In this low pH soft robot can keep its original shape. By sending the robot to the colon with a pH of around 8-8.5 soft robots can deform to its programmed shape to perform a function, for example, a biopsy. It can then be recovered from the body through the rectum.

It is a great suggestion. We added this discussion to the outlook section.

Changes in the manuscript:

Page 26: Adjusting our hydrogel to be triggered by milder pHs is another crucial factor for the practical applications.

6) Is it possible to achieve a specific deformation by locally dripping an acidic/alkaline solution on the nanocomposites in air environment?

We tried to achieve deformations by locally dripping the acidic or alkaline solution on the nanocomposites in the air. Results indicated that we cannot trigger considerable local shape changes by dripping the acidic or alkaline solution.

7) The authors describe the self-healing process of the material as follows in the article: “First, we studied the effect of time on the self-healing of GelWC and noticed a gradual enhancement of healing efficiency with time. After about 4 h GelWC showed full recovery, Fig. 2b.” and “The comparison between the results of mechanical testing on original physically crosslinked hydrogels of different formulation and their self-healed counterparts after 6 h also revealed the complete recovery of mechanical properties, or self-healing efficiency of 100%, containing at least 75 wt% of DMAPS (Fig. 2c).” My questions are:

1. During the self-healing process, the expressions about the healing results of 4 hours and 6 hours are basically the same, but we can clearly see the difference in the final healing effect in Fig. 2b. And the healing effect of 6 hours was significantly lower than that of 4 hours, and the author did not analyze the reasons for the decline in the article. It is also not indicated whether the proportion of its materials, environment and other factors are completely consistent during the healing process?

We tried to keep the healing process as consistent as possible for all the samples. We have done the healing experiment on at least 3 samples. Healing efficiencies based on tensile strength and elongation at break after 4 h of healing are 104.7 ± 2.5 % and 97.8 ± 4.1 %, respectively. Healing efficiencies based on tensile strength and elongation at break after 6 h of healing are 104.9 ± 5.8 % and 94.3 ± 3.5 %, respectively. As can be seen, the healing efficiencies are not significantly different. The variation in results can be due to the inconsistency in the thickness of the samples.

2. At the same time, in the description of “self-healing efficiency of 100%, containing at least 75 wt% of DMAPS (Fig. 2c).”, there is no clear determination of the content of DMAPS, if its content is in the self-healing process. If there is a change, is the healing effect of the material under different raw material ratios accurate? What is the obvious effect of the content of DMAPS on the self-healing effect of the material itself?

We examined the self-healing efficiency of different hydrogels with different weight ratios of DMAPS: MAA. Based on our results in Fig. 2c we found that the recovery for the samples with DMAPS: MAA ratio of 3:1 and 4:1 is nearly complete. Based on literature (J. Mater. Chem. B, 2019, 7, 1697--1707) the time we used for polymerization (1.5 h) is long enough to ensure the complete conversion of monomers to polymer so the ratio of comonomers in the final hydrogel should be the same as the initial ratio. The sample with a DMAPS: MAA weight ratio of 3:1 contains 75 wt% DMAPS with respect to the total monomer weight. We made it clear in the manuscript that this 75 wt% is with respect to the total monomer weight not the total weight of the hydrogel. DMAPS is the zwitterionic monomer responsible for dynamic electrostatic interactions and physical crosslinking between the oppositely charged groups as the main mechanism of self-healing in our hydrogel. That is why the amount of DMAPS is crucial for achieving the full recovery of mechanical properties after damage.

Changes in the manuscript:

Page 6: The comparison between the results of mechanical testing on original physically crosslinked hydrogels of different formulation and their self-healed counterparts after 6 h also revealed the complete recovery of mechanical properties, or self-healing efficiency of 100%, of the hydrogel containing at least 75 wt% of DMAPS with respect to the total monomer weight.

REVIEWERS' COMMENTS

Reviewer #1 (Remarks to the Author):

The authors have made considerable improvements to their manuscript after careful consideration of reviewer comments. In my opinion, almost all comments have been addressed satisfactorily. My remaining comment concerns analysis of SEM images presented in Fig S6.

The authors state on page 12 that "the gradient of microstructural anisotropy was less pronounced for films of 250 um thickness". In the SEM images, to me this appears to be the exact opposite.

Moreover, in relation to discussions surrounding CNC alignment, on page 12 the authors state that (regarding the image in Fig S6b) this image revealed layered structures with periodic spacing and spiral stacking of CNCs that are characteristic of chiral nematic assemblies. In my opinion, these features cannot be distinguished at all.

After addressing these small details, I feel that this manuscript would be appropriate for publication. Congratulations to the authors on a nice piece of work!

[Note from the Editor: Reviewer #1 was asked to look also over the response given to reviewer #2]

Yes I believe the authors have sufficiently addressed the concerns of reviewer 2.

Reviewer #3 (Remarks to the Author):

The authors have responded satisfactorily to most of my comments. However, I still see several minor issues in the article, which have to be improved before publication:

1)The comment (1) has not been modified thoroughly. The arrangement density of CNC

particles in Fig 7a is still inconsistent, please modify it.

2)In the response of comment (2), the authors used a reference to reinterpret an application of their material in soft robots. This reference can play a role in confirming, but hopes to find more references to support a broader explanation.

3)In the response of comment (4), the authors answered the property measurement of the material swelling in acidic and alkaline solutions for five minutes, which explains the doubts about whether the material can be functional in acidic and alkaline solutions. However, it is hoped that the description of the experiment and use limitations of materials at different pHs can be more accurate, such as in which pH ranges the material has almost no effect, and in which pHs it will be affected to varying degrees over time, etc. For example, the PH = 12 and PH = 3 of this measurement showed different results, indicating that the influence of acidic and alkaline environments on materials cannot be demonstrated uniformly. At the same time, based on this research on the acidity and alkalinity of the environment, the application scenarios anticipated by this material are objectively described in the article.

Programmable Nanocomposites of Cellulose Nanocrystals and Zwitterionic Hydrogels for Soft Robotics

*Rasool Nasser, Negin Bouzari, Junting Huang, Hossein Golzar, Sarah Jankhani, Xiaowu (Shirley) Tang, Tizazu H. Mekonnen, Amirreza Aghakhani and Hamed Shahsavan**

Response letter to reviewers

NCOMMS-23-10872-T

We are grateful to the reviewers for their detailed comments that helped us improve our manuscript significantly. We have addressed the comments from each reviewer point-by-point in this response letter. For clarity purposes, we have marked our response to the reviewer's comment in blue. We have revised the main text accordingly and highlighted the changes in this letter and the main text in yellow to facilitate the revision. We hope our revisions address the reviewers' questions and suggestions and that the revised manuscript meets the standards for publication in Nature Communication.

REVIEWER COMMENTS

Reviewer #1 (Remarks to the Author):

The authors have made considerable improvements to their manuscript after careful consideration of reviewer comments. In my opinion, almost all comments have been addressed satisfactorily. My remaining comment concerns analysis of SEM images presented in Fig S6.

The authors state on page 12 that "the gradient of microstructural anisotropy was less pronounced for films of 250 um thickness". In the SEM images, to me this appears to be the exact opposite.

Moreover, in relation to discussions surrounding CNC alignment, on page 12 the authors state that (regarding the image in Fig S6b) this image revealed layered structures with periodic spacing and spiral stacking of CNCs that are characteristic of chiral nematic assemblies. In my opinion, these features cannot be distinguished at all.

After addressing these small details, I feel that this manuscript would be appropriate for publication. Congratulations to the authors on a nice piece of work!

We revised the discussion of Fig. S6. For Fig. S6a, we focused on the similar morphology of hydrogels prepared in a cell with 500 μm and 250 μm thicknesses. In Fig. S6b, we attributed the morphology to the polydomain microstructure of CNC by referring to a published literature in which authors saw a similar morphology in a similar system (Nature Communications, (2019)10:510)

Changes in the manuscript:

Page 12: The gradient of microstructural anisotropy along the thickness can be observed even after reducing the sample thickness to 250 μm (Supplementary Fig. 6a). To compare the microstructure, the SEM image of the cross-section of IGel with 500 μm is presented in Supplementary Fig. 6b. Layered structures closer to the substrate farther from the shear resemble randomly oriented domains of chiral nematic CNC aggregates reported in the literature.⁷⁰ Further investigation is required to identify the exact nature of the phase and alignment.

Reviewer #3 (Remarks to the Author):

The authors have responded satisfactorily to most of my comments. However, I still see several minor issues in the article, which have to be improved before publication:

1)The comment (1) has not been modified thoroughly. The arrangement density of CNC particles in Fig 7a is still inconsistent, please modify it.

We revised Fig. 7a.

Changes in the manuscript:

Page 25: Fig. 7

2)In the response of comment (2), the authors used a reference to reinterpret an application of their material in soft robots. This reference can play a role in confirming, but hopes to find more references to support a broader explanation.

We added more details on the safe interaction of soft robots with the human body to the manuscript with more references.

Changes in the manuscript:

Page 6: The similarity of the elastic modulus of soft robots and soft body tissues is essential for the non-invasive interaction of them with the body as well as the manipulation of soft objects, like

tissue cells. The elastic modulus of our hydrogel is around 30 kPa which is in the range of some soft body tissues (such as Uterus tissue ~ 2-250 kPa and Liver tissue ~ 10 kPa) and smaller than some others (such as Kidney tissue ~ 90-180 kPa and Small intestinal tissue ~ 2500-5500 kPa) ensuring the safe and non-invasive interactions.

3) In the response of comment (4), the authors answered the property measurement of the material swelling in acidic and alkaline solutions for five minutes, which explains the doubts about whether the material can be functional in acidic and alkaline solutions. However, it is hoped that the description of the experiment and use limitations of materials at different pHs can be more accurate, such as in which pH ranges the material has almost no effect, and in which pHs it will be affected to varying degrees over time, etc. For example, the PH = 12 and PH = 3 of this measurement showed different results, indicating that the influence of acidic and alkaline environments on materials cannot be demonstrated uniformly. At the same time, based on this research on the acidity and alkalinity of the environment, the application scenarios anticipated by this material are objectively described in the article.

Because our hydrogel does not swell very much at low pHs, it just experiences very low-speed degradation due to the hydrolysis in the acidic medium. We examined the stability of our hydrogels in acidic mediums for up to a few weeks and did not observe any noticeable change. At high pHs (above the pKa of methacrylic acid) though, due to internal pressure exerted by the absorbed water, hydrogel becomes stiff and the same time more susceptible to rupture. To use our hydrogel in a medium with a certain pH range in a real-world application, an elaborate study should be performed on the effect of its long-time exposure to that pH range. We added this suggestion to the outlook of our manuscript.

Changes in the manuscript:

Page 25: In this work, we have investigated the shape-change properties in two extreme pH environments to expedite actuation. However, the pKa of the methacrylic acid dictates that the deformation of the system occurs at pH levels above 4.6 in principle. Further details are required to determine the exact pH profile that is effective in inducing shape changes. Our system holds the potential for therapeutic applications in body organs with high native pH levels, as well as the ability to tolerate acidic pH environments. The bladder serves as an example of such a favorable environment.⁸⁸ In addition, the durability of our hydrogel construct along with their on-demand degradability needs to be alleviated. To employ our hydrogel in a medium with a certain pH range in a real-world application, an elaborate study should be performed on the effect of its long-time exposure to that pH range.